# SEQUENTIAL BAYESIAN CONTINUAL LEARNING WITH META-LEARNED NEURAL NETWORKS

## ABSTRACT

In the present era of deep learning, continual learning research is mainly focused on mitigating forgetting when training a neural network with stochastic gradient descent (SGD) on a non-stationary stream of data. On the other hand, there is a wealth of research on sequential learning in the more classical literature of statistical machine learning. Many models in this literature have sequential Bayesian update rules that yield the same learning outcome as the batch training, i.e., they are completely immune to catastrophic forgetting. However, they suffer from underfitting when modeling complex distributions due to their weak representational power. In this work, we introduce a general meta-continual learning (MCL) framework that combines neural networks' strong representational power and simple statistical models' robustness to forgetting. In our framework, continual learning takes place only in a statistical model in the embedding space via a sequential Bayesian update rule, while meta-learned neural networks bridge the raw data and the embedding space. Since our approach is domain-agnostic and model-agnostic, it can be applied to a wide range of problems and easily integrated with existing model architectures. Compared to SGD-based MCL methods, our approach demonstrates significantly improved performance and scalability.

## 1 INTRODUCTION

Continual learning (CL), acquiring new knowledge or skills without forgetting existing ones, is an essential ability of intelligent agents. Despite recent advances in deep learning, CL remains a significant challenge, and there is a compelling reason for this difficulty. Knoblauch et al. (2020) rigorously proves that, in general, CL is an NP-hard problem. This implies that building a universal CL algorithm effective for all scenarios is impossible as long as P$\neq$NP. To effectively tackle a CL problem, there must be some inherent structure in the problem, and the algorithm should possess prior knowledge to leverage it. Even in the case of humans, our CL abilities are specialized for specific tasks, such as learning new faces, and are not as effective for others, like memorizing random digits. This specialization results from the evolutionary process that has optimized our CL abilities for survival and reproduction.

From this perspective, meta-continual learning (MCL) emerges as a highly promising avenue of research. Rather than manually crafting a CL algorithm based on human knowledge, MCL aims to meta-learn the CL ability in a data-driven manner. Therefore, it is often described as *learning to continually learn*. Like meta-learning, which encompasses multiple learning episodes, the MCL problem setting involves numerous CL episodes. Each of these episodes can be likened to the lifecycle of an organism in an evolutionary process. MCL also follows the bi-level optimization structure of meta-learning: in the inner loop, a CL algorithm produces a model trained on a CL episode, while in the outer loop, the CL algorithm is optimized across multiple episodes.

Since stochastic gradient descent (SGD) is the primary optimization method for neural networks, most CL research focuses on mitigating forgetting the previous knowledge when training neural networks on a non-stationary data stream. As such, several approaches in MCL (Javed & White, 2019; Beaulieu et al., 2020) incorporate SGD as the primary update rule in the inner loop.

Meanwhile, the bi-level optimization structure of MCL offers the flexibility to combine meta-learned neural networks with inner update rules other than SGD. In this context, the sequential Bayesian update stands out as the most promising candidate, providing an ideal framework for updating a

knowledge state. While there have been a significant number of CL approaches based on the idea of updating the posterior belief of neural network parameters (Kirkpatrick et al., 2016; Zenke et al., 2017; Chaudhry et al., 2018; Nguyen et al., 2018; Farquhar & Gal, 2019), various approximations are necessary to ensure computational tractability, which sets them apart from the ideal Bayesian update. On the other hand, we bring the Fisher-Darmois-Koopman-Pitman theorem (Fisher, 1934; Darmois, 1935; Koopman, 1936; Pitman, 1936) into the scope to point out that the exponential family is the only family of distributions that are capable of efficient and lossless sequential Bayesian update (more precise description in §2.2). Instead of dealing with the intractable posterior of complex neural networks, we consider the sequential Bayesian inference of simple statistical models that inherently come with an exponential family posterior, yielding a result identical to batch inference. While these models are immune to catastrophic forgetting by design, they are often too simple for modeling complex, high-dimensional data. Fortunately, the MCL setting offers meta-learned neural networks that can work as bridges between complex real-world and a streamlined embedding space where the statistical models can thrive.

We distill this idea of combining simple statistical models and meta-learned neural networks into a general MCL framework, which we call *Sequential Bayesian Meta-Continual Learning (SB-MCL)* Since SB-MCL is domain-agnostic and model-agnostic, it can be applied to a wide range of problem domains and integrated with existing model architectures with minimal modifications. SB-MCL encompasses several prior works (Banayeeanzade et al., 2021; Snell et al., 2017; Harrison et al., 2018) as special cases and supports both supervised and unsupervised learning scenarios.

## 2 BACKGROUND

### 2.1 META-CONTINUAL LEARNING

We start by describing the problem setting of MCL. We denote an example $(x, y)$ where $x$ is an input variable, and $y$ is an output variable, assuming a supervised setting by default. One can replace $(x, y)$ with $x$ for unsupervised learning settings. A CL episode $(\mathcal{D}, \mathcal{E})$ consists of a training stream $\mathcal{D} = ((x_t, y_t))_{t=1}^T$ and a test set $\mathcal{E} = \{(\tilde{x}_n, \tilde{y}_n)\}_{n=1}^N$. The training stream is an ordered sequence of length $T$, and its examples can only be accessed sequentially and cannot be accessed more than once. It is assumed to be non-stationary and typically constructed as a concatenation of $K$ distinct *task* streams. Naively training a neural network on such a non-stationary stream with SGD will result in catastrophic forgetting of the knowledge from the previous part of the stream. The test set consists of examples of the tasks appearing in the training stream, such that the model needs to retain knowledge of all the tasks to obtain a high score in the test set. In MCL, multiple CL episodes are split into meta-training and meta-test sets. During the meta-training phase, a CL algorithm is optimized across multiple episodes in the meta-training set to produce a competent model from a training stream. The algorithm's CL capability is then measured on the meta-test set. Note that meta-training and meta-test sets typically do not share any underlying tasks since the meta-test set aims to measure the learning capability, not the knowledge of specific tasks that appear during meta-training.

We emphasize that MCL and CL are two distinct problem settings with different underlying assumptions and objectives. While the goal of CL is to produce a model for a single episode, MCL aims to learn a CL algorithm that can produce models for various CL episodes. There are numerous other approaches that combine meta-learning and CL (Finn et al., 2019; Riemer et al., 2019; Jerfel et al., 2019; Gupta et al., 2020; to name a few) but differ in problem settings.

### 2.2 SEQUENTIAL BAYESIAN LEARNING AND THE EXPONENTIAL FAMILY

The Bayes rule offers a principled way to update knowledge incrementally by using the posterior at the previous time step as the prior for the current time step (Bishop, 2006; Murphy, 2022), i.e., $p(z|x_{1:t}) \propto p(x_t|z)p(z|x_{1:t-1})$. Therefore, the Bayesian perspective has been widely adopted in CL research (Kirkpatrick et al., 2016; Zenke et al., 2017; Chaudhry et al., 2018; Nguyen et al., 2018; Farquhar & Gal, 2019). However, prior works have focused on sequentially updating the posterior of neural network parameters, which are generally intractable to compute. Therefore, they must rely on various approximations, resulting in a wide gap between the ideal Bayesian update and reality.

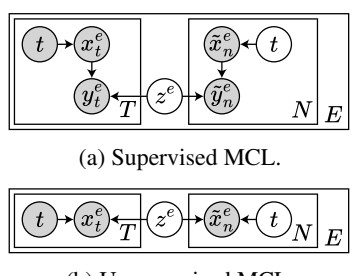

(a) Supervised MCL.

(b) Unsupervised MCL.

Figure 1: Graphical models of MCL. For each episode $e$, examples $(x_t^e, y_t^e)$ (or just $x_t^e$) are produced conditioned on the time step $t$ and the episode-wise latent variable $z^e$.

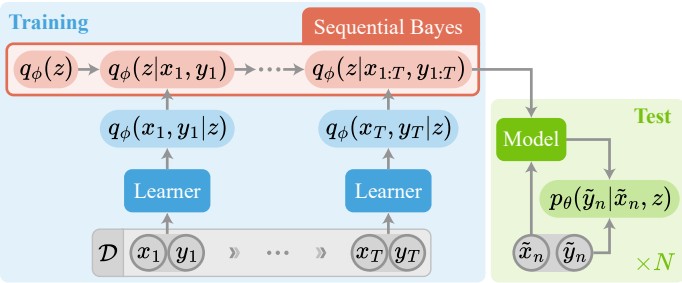

Figure 2: Schematic diagram of our SB-MCL in a single supervised CL episode. In SB-MCL, CL is formulated as the sequential Bayesian update of the variational posterior of the latent variable $z$. Since the neural network components (the learner and the model) are meta-learned and remain fixed during training, they are protected from catastrophic forgetting.

Then, what kind of models are suitable for efficient sequential Bayesian updates? According to the Fisher-Darmois-Koopman-Pitman theorem (Fisher, 1934; Darmois, 1935; Koopman, 1936; Pitman, 1936), *the exponential family is the only family of distributions where the dimension of the sufficient statistic remains fixed, regardless of the number of examples.* This theorem has significant implications for CL; if the model's posterior is not a member of the exponential family (as in the case of neural networks) and does not have a large enough memory system to store the ever-growing sufficient statistics, forgetting becomes inevitable. From this perspective, employing a replay buffer (Lopez-Paz & Ranzato, 2017; Chaudhry et al., 2019) is an approach that aids in partially preserving sufficient statistics.

On the flip side, the theorem suggests an alternative approach; by embracing an exponential family distribution, we can store sufficient statistics within a fixed dimension, enabling efficient sequential Bayesian updates without any compromises (see also Banayeeanzade et al. (2021)). Although the exponential family's expressivity is limited, this challenge can be effectively addressed in MCL settings by employing meta-learned neural networks to translate complex data into simplified embeddings and vice versa.

## 3 SEQUENTIAL BAYESIAN META-CONTINUAL LEARNING

### 3.1 THE VARIATIONAL BOUND

Fig. 1 shows the graphical models of our MCL settings. In both supervised and unsupervised settings, there are $E$ CL episodes. Each CL episode $e$ has a training stream $\mathcal{D}^e$ of length $T$ and a test set $\mathcal{E}^e$ of size $N$. In supervised CL settings (Fig. 1a), each example is a pair of input $x$ and target $y$, and the goal is to model the conditional probability $p(y|x)$. In unsupervised settings (Fig. 1b), an example is simply $x$, without distinction of input and target, and the goal is to model $p(x)$. For each CL episode $e$, we assume an episode-specific latent variable $z^e$ that governs the entire episode. The training stream's non-stationarity, a key characteristic of CL, is expressed by the time variable $t$ affecting the generation of $x$. In practice, the training stream is often constructed by concatenating multiple *task* streams, each of which is a stationary stream sampled from a distinct task distribution. Note that $z^e$ is shared by all examples inside an episode regardless of the tasks they belong to. Under this framework, the CL process is to sequentially refine the belief state of $z^e$.

The objective is to maximize the (conditional) log-likelihood of the test set $\mathcal{E}$ after continually learning from the training stream $\mathcal{D}$ (we will now omit the superscript $e$ for brevity). Assuming a *model* parameterized by $\theta$, this objective can be summarized as $\log p_\theta(\tilde{y}_{1:N}|\tilde{x}_{1:N}, \mathcal{D}) = \sum_{n=1}^{N} \log p_\theta(\tilde{y}_n|\tilde{x}_n, \mathcal{D})$ in supervised settings and as $\log p_\theta(\mathcal{E}|\mathcal{D}) = \log p_\theta(\tilde{x}_{1:N}|\mathcal{D}) = \sum_{n=1}^{N} \log p_\theta(\tilde{x}_n|\mathcal{D})$ in unsupervised settings, where $\tilde{x}_*$ and $\tilde{y}_*$ are the test data in $\mathcal{E}$, and $\theta$ is the model parameter. Since computing these objectives is generally intractable due to the latent variable $z$, we instead introduce a variational distribution $q_\phi$ parameterized by $\phi$ and derive the variational

lower bounds. For the supervised and unsupervised cases, the bounds are derived as follows:

$$\log p_\theta(\tilde{y}_{1:N}|\tilde{x}_{1:N}, \mathcal{D}) = \log p_\theta(\tilde{y}_{1:N}|\tilde{x}_{1:N}, x_{1:T}, y_{1:T})$$

$$\geq \mathbb{E}_{z \sim q_\phi(z|\mathcal{D})} \left[ \sum_{n=1}^{N} \log p_\theta(\tilde{y}_n|\tilde{x}_n, z) + \sum_{t=1}^{T} \log p_\theta(y_t|x_t, z) \right] - D_{\mathrm{KL}}\left(q_\phi(z|\mathcal{D}) \,\|\, p_\theta(z)\right) - \underbrace{\log p_\theta(\mathcal{D})}_{\mathrm{const.}} \quad (1)$$

$$\log p_\theta(\tilde{x}_{1:N}|\mathcal{D}) = \log p_\theta(\tilde{x}_{1:N}|x_{1:T})$$

$$\geq \mathbb{E}_{z \sim q_\phi(z|\mathcal{D})} \left[ \sum_{n=1}^{N} \log p_\theta(\tilde{x}_n|z) + \sum_{t=1}^{T} \log p_\theta(x_t|z) \right] - D_{\mathrm{KL}}\left(q_\phi(z|\mathcal{D}) \,\|\, p_\theta(z)\right) - \underbrace{\log p_\theta(\mathcal{D})}_{\mathrm{const.}} \quad (2)$$

Note that Garnelo et al. (2018b) employ a similar derivation for neural processes, with an objective akin to our supervised settings. However, they introduce an additional approximation in the middle, resulting in an improper bound (Volpp et al., 2021; Le et al., 2018). In contrast, we refrain from making such an approximation and instead derive a proper bound. For a more detailed explanation, please refer to Appendix A.

## 3.2 Sequential Bayesian Update of the Variational Posterior

In Eq. 1 and 2, the CL process is abstracted inside the variational posterior $q_\phi(z|\mathcal{D})$, which should be obtained through sequential Bayesian updates:

$$q_\phi(z|x_{1:t}, y_{1:t}) \propto q_\phi(x_t, y_t|z)q_\phi(z|x_{1:t-1}, y_{1:t-1}), \quad q_\phi(z|x_1, y_1) \propto q_\phi(x_1, y_1|z)q_\phi(z) \quad (3)$$

$$q_\phi(z|x_{1:t}) \propto q_\phi(x_t|z)q_\phi(z|x_{1:t-1}), \quad q_\phi(z|x_1) \propto q_\phi(x_1|z)q_\phi(z) \quad (4)$$

where Eq. 3 and 4 are respectively for supervised and unsupervised CL. As previously explained in §2.2, the Fisher-Darmois-Koopman-Pitman theorem implies that only exponential family distributions can perform such updates without consistently increasing the memory and compute requirement proportional to the number of examples. This property makes them ideal candidates for our variational posterior. In the following, we will assume the posterior as a factorized Gaussian distribution, but similar derivations apply to other exponential family distributions.

First, we define the variational prior $q_\phi(z) = \mathcal{N}(z; \mu_0, \Lambda_0^{-1})$. We then employ a neural network parameterized by $\phi$ to produce $q_\phi(x_t, y_t|z)$ or $q_\phi(x_t|z)$, which we refer to as the *learner*. In the case of the Gaussian variational posterior, which has the form of $\mathcal{N}(z; \mu_t, \Lambda_t^{-1})$, the learner outputs $\hat{z}_t$ and $P_t$ for each $(x_t, y_t)$, where $P_t$ is a diagonal matrix. We consider $\hat{z}_t$ a noisy observation of $z$ with a Gaussian noise of precision $P_t$ similarly to Volpp et al. (2021), i.e., $q_\phi(x_t, y_t|z) = \mathcal{N}(\hat{z}_t; z, P_t^{-1})$. This allows an efficient sequential update rule for the variational posterior as follows (Bishop, 2006):

$$\Lambda_t = \Lambda_{t-1} + P_t, \quad \mu_t = \Lambda_t^{-1}\left(\Lambda_{t-1}\mu_{t-1} + P_t\hat{z}_t\right). \quad (5)$$

## 3.3 Meta-Training and Meta-Testing

Let us begin by explaining the meta-test phase to provide an overall understanding of how a fully trained SB-MCL operates. Fig. 2 illustrates SB-MCL functions in a supervised CL episode during the meta-test phase. As each example $(x_t, y_t)$ in the training stream $\mathcal{D}$ becomes available sequentially, the learner processes it into $q_\phi(x_t, y_t|z)$, which is then incorporated into the variational posterior $q_\phi(z|x_{1:t}, y_{1:t})$ by Eq. 3. After training, the final posterior $q_\phi(z|x_{1:T}, y_{1:T})$ is passed on to the test phase. During testing, the model produces outputs conditioned on the test input $\tilde{x}_n$ and $z$. It would be ideal if we could analytically compute $\mathbb{E}_{z \sim q_\phi(z|x_{1:T}, y_{1:T})}[p_\theta(\tilde{y}_n|\tilde{x}_n, z)]$. But if this is not the case, we may sample multiple $z$'s from $q_\phi(z|x_{1:T}, y_{1:T})$ and ensemble the outputs conditioned on each $z$, or feed the maximum a posteriori estimation $z_{\mathrm{MAP}}$ to the model. A similar procedure is performed in unsupervised settings as well.

There are some crucial differences compared to SGD-based MCL. First, while SGD-based methods execute the model on the training examples to produce the loss and update its parameters with gradient descent, SB-MCL utilizes separate learner and sequential Bayesian updates to learn the training stream. Note that although we conceptually introduce a separate learner, the learner and the model may share components to promote parameter efficiency or generalization. Additionally, SB-MCL does not involve any gradient descent during training; the learner performs only the forward

passes to process the training examples for sequential Bayesian updates. However, this necessitates a meta-training phase, since the learner and the model must be trained beforehand.

During the meta-training phase, the model and the learner are meta-updated to maximize the Eq. 1 or 2 for each CL episode. For each episode, the first step is to perform the inner loop: going through the training stream to compute the variational posterior $q_\phi(z|\mathcal{D})$. In contrast to SGD-based MCL, our approach does not need to process the training stream sequentially. If all the training examples are available, which is generally true during meta-training, we can feed them to the learner in parallel and combine the results with a batch inference rule instead of the sequential update rule. For the Gaussian posterior, we can use the following formula instead of Eq. 5 to produce an identical result:

$$\Lambda_T = \sum_{t=0}^{T} P_t, \quad \mu_T = \Lambda_T^{-1} \sum_{t=0}^{T} P_t \hat{z}_t \tag{6}$$

Compared to SGD-based approaches requiring forward-backward passes for each example sequentially, the meta-training of our approach can benefit from parallel processors such as GPUs or TPUs.

Once the variational posterior $q_\phi(z|\mathcal{D})$ is obtained, we use Monte Carlo approximation for the expectation w.r.t. $q_\phi(z|\mathcal{D})$ (Kingma & Welling, 2014). For our Gaussian posterior, we utilize the reparameterization trick (Kingma & Welling, 2014) to sample $z$ that allows backpropagation.

$$z = \mu_T + \Lambda_T^{-1/2}\epsilon, \quad \epsilon \sim \mathcal{N}(0, I) \tag{7}$$

Conditioned on this $z$, we run the model on the training and test examples to compute the first term in Eq. 1 or 2. This term encourages the cooperation between the model and the learner to increase the likelihood of the data. The second term is the Kullback-Leibler (KL) divergence between the variational posterior $q_\phi(z|\mathcal{D})$ and the prior $p_\theta(z)$, which can be regarded as a regularization term. We set the prior to be the same exponential family distribution, e.g., the unit Gaussian for the Gaussian posterior, which enables an analytical computation of the KL divergence. Finally, the last term $\log p_\theta(\mathcal{D})$ is a constant that can be ignored for optimization purposes.

After Eq. 1 or 2 is computed for an episode or a batch of episodes, we perform a meta-update on the model and the learner with an SGD algorithm, backpropagating through the entire episode. Unlike existing SGD-based MCL methods (Javed & White, 2019; Beaulieu et al., 2020), we do not need to calculate any second-order gradients, which is a significant advantage for scalability.

## 3.4 SPECIAL CASES OF SB-MCL

**GeMCL (Banayeeanzade et al., 2021).** GeMCL can be regarded as a specific instance of our framework in the image classification domain. It utilizes a meta-learned neural network encoder to extract an embedding vector for each image. During the training process, it maintains a Gaussian posterior for each class in the embedding space. Each Gaussian posterior is updated by the sequential Bayesian update rule as each example belonging to the corresponding class becomes available. These Gaussians collectively form a Gaussian mixture model (GMM) within the embedding space. At test time, each test image is converted into an embedding vector by the same encoder, and a class is predicted by inferring the mixture component of the GMM. To view GeMCL as an instance of SB-MCL, we can consider the encoder as serving two roles: one as the learner and the other as a component of the model. During training, the encoder is used as the learner to update the posterior $q_\phi(z|x_{1:t}, y_{1:t})$ where $z$ is the parameters of the GMM. At test time, the encoder transforms the test inputs into embeddings as a model component, and the GMM classifies the embeddings with its parameters produced from the training phase. Banayeeanzade et al. (2021) also propose an MAP variant, which simply produces $p_\theta(\tilde{y}_n|\tilde{x}_n, z_{\text{MAP}})$ as the output. This variant has simpler computation without significant performance drops.

**Prototypical Networks (Snell et al., 2017).** While GeMCL is a special case of SB-MCL, it can also be seen as a generalization of the Prototypical Network (PN), which was originally proposed as a meta-learning approach for few-shot classification. Therefore, PN also falls under the SB-MCL family. While GeMCL takes a fully Bayesian approach, PN simply averages the embeddings of each class to construct a prototype vector. Since the average operation can be performed sequentially, PN can be readily applied to MCL settings. We can simplify GeMCL to PN by assuming isotropic Gaussian posteriors and an uninformative prior (Banayeeanzade et al., 2021).

Table 1: Summary of the special cases of SB-MCL

| Method | Model structure | Learner structure | $q_\phi(z|\mathcal{D})$ |
|---|---|---|---|
| GeMCL | $x$-encoder + GMM | $x$-encoder (shared) | Per-class Gaussian |
| PN | $x$-encoder + GMM | $x$-encoder (shared) | Per-class isotropic Gaussian |
| ALPaCA | $x$-encoder + linear model | $x$-encoder (shared) | Matrix normal |
| SB-MCL (supervised) | $x$-encoder + $y$-decoder | $xy$-encoder | Factorized Gaussian |
| SB-MCL (unsupervised) | Deep generative model | $x$-encoder | Factorized Gaussian |

**ALPaCA (Harrison et al., 2018).** Originally proposed as a meta-learning approach for online regression problems, ALPaCA attaches a linear model on top of a meta-learned neural network encoder, symmetrical to PN or GeMCL that attaches a GMM for classification. In ALPaCA, the latent variable $z$ is the weight matrix of the linear model, whose posterior is assumed to have the matrix normal distribution. Due to the similar streaming settings of online and continual learning, we can apply ALPaCA to MCL regression settings with minimal modifications.

**Unlocking General Domains with Generic SB-MCL Architectures.** All these prior works share a similar architecture: a meta-learned encoder followed by a simple statistical model. This configuration can be ideal if the output type is suitable for the statistical model since it allows analytic computation of $\mathbb{E}_{z \sim q_\phi(z|\mathcal{D})}[p_\theta(\tilde{y}_n|\tilde{x}_n, z)]$ without expansive Monte Carlo approximation. However, it is hard to apply such architectures to domains with more complex output formats or unsupervised settings where the output variable does not exist. Our SB-MCL offers a solution to this problem, allowing the combination of simple statistical models and general deep learning architecture. Since the only modification required for the model is to accept additional input $z$, we can apply SB-MCL to almost any existing model architectures or domains. The specific instantiations of SB-MCL are summarized in Table 1 for better understanding.

## 4 RELATED WORK

**SGD-Based MCL Approaches.** In contrast to the simple statistical models of our special cases in §3.4, OML (Javed & White, 2019) employs a small multi-layer perceptron (MLP) with MAML (Finn et al., 2017) on top of a meta-learned encoder. MAML is a meta-learning approach that optimizes the initial parameters of a model by meta-gradient descent, computing the second-order gradient through the inner loop. In the inner loop of OML, the encoder remains fixed while the MLP is updated by sequentially learning each training example via SGD. After training the MLP in the inner loop, the entire model is evaluated on the test set to produce the meta-loss. Then, the gradient of the meta-loss is computed w.r.t. the encoder parameters and the initial parameters of the MLP to update them. Inspired by OML, ANML (Beaulieu et al., 2020) is another MCL method for image classification that introduces a separate component called neuromodulatory network. Its sigmoid output is multiplied to the encoder output to adaptively gate some features depending on the input.

**Neural Processes (Garnelo et al., 2018a;b).** While motivated by different objectives, intriguing similarities can be identified between the supervised version of SB-MCL (Eq. 1) and the neural process (NP) literature. NP was initially proposed to solve the limitations of Gaussian processes, such as the computational cost and the difficulties in the prior design. It can also be considered a meta-learning approach that learns a functional prior and has been applied as a solution to the meta-learning domain (Gordon et al., 2019). Since NPs are rooted in stochastic processes, one of their primary design considerations is exchangeability: the model should produce the same result regardless of the order of the training data. NPs typically comprise an encoder and a decoder, analogous to the learner and the model in our framework. To achieve exchangeability, NPs independently encode each example and aggregate them into a single variable with a permutation-invariant operation, such as averaging, and pass it to the decoder. While our sequential Bayesian update of an exponential family posterior is initially inspired by the Fisher-Darmois-Koopman-Pitman theorem, it also ensures exchangeability. Volpp et al. (2021) propose an aggregation scheme for NPs based on Bayesian principles and even suggest the possibility of sequential update, but they do not connect it to CL. To the best of our knowledge, the only connection between NPs and MCL is CNAP (Requeima et al., 2019), but it is a domain-specific architecture designed for image classification.

# 5 EXPERIMENTS

We now demonstrate the efficacy of our framework on multiple domains, including both supervised and unsupervised tasks. We also provide PyTorch (Paszke et al., 2019) code, ensuring the reproducibility of all experiments. Due to page limitations, we present only the most essential information; for further experimental details, please refer to the code.

## 5.1 METHODS

**The SB-MCL Family.** We test the special cases of SB-MCL in Table 1 for their respective domains, i.e., GeMCL for image classification, ALPaCA for simple regression, and the generic supervised and unsupervised variants for others. GeMCL and ALPaCA support the analytic calculation of posterior predictive distribution during testing. For the generic cases, we impose 512-dimensional factorized Gaussian on $q_\phi(z|\mathcal{D})$ and sample $z$ five times to approximate $\mathbb{E}_{z \sim q_\phi(z|\mathcal{D})}[p_\theta(\tilde{y}_n|\tilde{x}_n, z)]$. In the appendix, we also test its MAP variant that simply produces $p_\theta(\tilde{y}_n|\tilde{x}_n, z_{\mathrm{MAP}})$.

**Baselines.** Due to its simplicity and generality, we test OML (Javed & White, 2019) as a representative baseline of SGD-based MCL. Although it was originally proposed for classification and simple regression, we can implement the core idea of having a MAML MLP block working in the embedding space to other domains. For models with encoder-decoder architectures, we insert a MAML MLP between the meta-learned encoder and decoder. For comparison, we also test vanilla MAML (Finn et al., 2017), which updates the entire model in the inner loop and meta-optimizes its initialization. Additionally, we test Reptile (Nichol et al., 2018), which is a first-order approximation of MAML that does not involve expensive computation of second-order derivatives. We also compare a reptile variant of OML, which replaces the MAML MLP with a Reptile MLP. The results of MAML and Reptile, which are generally worse than OML, are provided in Appendix C.

**Without Meta-Learning.** Although our work is an MCL work, a significant number of non-meta-CL methods have been proposed. To provide a reference point to them, we report the standard and online learning scores, which are generally considered the upper bound of CL and online CL performance (Zenke et al., 2017; Farajtabar et al., 2020). For standard learning, we train a model from scratch for an unlimited number of steps with mini-batches uniformly sampled from the entire training stream. Since the model usually overfits to the training set, we report the best test score achieved during training. For online learning, we randomly shuffle the training stream to be a stationary stream, train a model from scratch for one epoch, and measure the final test score. Note that MCL methods can outperform standard and online learning since they can utilize a large meta-training set.

## 5.2 BENCHMARKS

As a popular meta-learning or MCL dataset, Omniglot (Lake et al., 2015) comprises 1,623 classes with 32K images. However, since there are only 20 images for each class, it can only test few-shot settings. The small number of examples also seems to cause meta-overfitting, which becomes especially severe when we use it for more complex tasks other than classification. Other popular datasets such as CIFAR-100 (Krizhevsky, 2009) and MiniImageNet (Vinyals et al., 2016; Deng et al., 2009) consist of several hundred images per class but offer only 100 classes.

As an alternative, we suggest utilizing the CASIA Chinese handwriting dataset (Liu et al., 2011) in MCL. It contains 3.9M handwriting images of 7,356 classes, which are mostly Chinese characters. While the number of examples varies across classes, the minimum count per class stands at 279. With its richness in both class diversity and examples per class, it enables comprehensive evaluations in many-shot scenarios and substantially rules out meta-overfitting issues. Except for the sine regression, the following benchmarks are built upon the CASIA dataset, using each class as a task.

**Classification.** We conduct image classification experiments with the Omniglot and CASIA datasets, following the setups of Banayeeanzade et al. (2021). Using the CASIA dataset, we also analyze the performances in many-shot settings. All the methods share the same CNN encoder while using different output mechanisms, i.e., GMM for GeMCL and PN, and MLP for others. GeMCL is compared as an instantiation of SB-MCL.

**Sine Regression.** Inspired by the synthetic sine wave regression problem from Javed & White (2019), we design a more challenging variant. We test ALPaCA as an instance of SB-MCL. A sine wave, denoted as a function $\omega(\tau)$, is characterized by amplitude $A$, frequency $\nu$, and phase $\psi$. We define the target signal $y$, as the set of values of the sine wave at 50 fixed time points: $y = [\omega(\tau_1), \ldots, \omega(\tau_{50})]$. In each task, all $y$ values share the same frequency and phase, while varying in amplitudes. We build the input $x$ by corrupting $y$ with a phase shift and Gaussian noise. The amount of phase shift is randomly sampled and shared for each task. We use the same MLP encoder for all the methods, followed by method-specific output modules: a linear model for ALPaCA, and another MLP for others.

**Image Completion.** Compared to the sine regression problem, this is a significantly more challenging high-dimensional regression problem. In this problem, $x$ and $y$ are an image's top and bottom halves, and each class is defined as a task. We use convolutional encoder-decoder architecture for the model. In the case of SB-MCL, we use the generic architecture for supervised settings and introduce another convolutional encoder for the learner, which produces $q_\phi(x, y|z)$ from a full training image. The model's decoder is also slightly modified to take the concatenation of the encoder's output and $z$ as input.

**Rotation Prediction.** In this problem, a model is given a randomly rotated image $x$ and tasked to predict the rotation angle $y$. Although the rotation angle is not high-dimensional, we use the generic supervised SB-MCL architecture as in the image completion benchmark. This is due to the objective function, which is defined as $1 - \cos(y - \hat{y})$ and cannot be used for analytically computing the posterior of the linear model in ALPaCA. For the model architecture, we use a convolutional encoder followed by an MLP output module. For the learner in SB-MCL, we share the same encoder in the model for encoding $x$ and introduce a new MLP to encode $y$. These two encoders' output is concatenated and fed to another MLP to produce $q_\phi(x, y|z)$.

**Deep Generative Modeling.** Lastly, we evaluate unsupervised learning performances with two types of deep generative models: variational autoencoder (VAE; Kingma & Welling, 2014) and denoising diffusion probabilistic models (DDPM; Ho et al., 2020). We use a simple convolutional encoder-decoder architecture for VAE and a U-Net encoder-decoder architecture for DDPM following Ho et al. (2020). In SB-MCL, we use a separate convolutional encoder for the learner, and $z$ is injected into the model by concatenating it with the decoder's input. For OML, we replace the encoder's last MLP and the decoder's first MLP with MAML MLP.

**Evaluation Scheme.** In all MCL experiments, we meta-train the methods in 10-task 10-shot settings. We primarily evaluate their performance in a meta-test set with the same task-shot setting, while also measuring the extrapolation capability on other meta-testing setups, which vary in the number of tasks or shots. The hyperparameters are tuned to maximize the performance in the 10-task 10-shot settings. For each MCL experiment, we report the average and the standard deviation of five runs. Within each MCL run, we calculate the average score from 512 CL episodes sampled from the meta-test set. For standard and online learning, which involve only one episode, we sample an episode from the meta-test set, train the model on the training set, and measure the test score. Subsequently, we provide the average and standard error of the mean for ten runs.

## 5.3 RESULTS

We present our main experimental results in Table 2 and Fig. 3. For qualitative examples and more extensive results, please refer to Appendix B and C. We mainly compare the SB-MCL family against OML, an SGD-based baseline, and standard learning, the upper bound of CL without meta-learning. The SB-MCL family consistently outperforms all the other approaches by a significant margin, demonstrating the efficacy of our framework. In the following, we discuss several important characteristics of our framework that can be observed in the experiments.

**Robustness to Many-Shot Settings.** Interestingly, the performance of SGD-based approaches can degenerate as we increase the number of shots per task (the 10-task X-shot plots in Fig. 3). This may seem counterintuitive, as providing more information about a task should generally be beneficial. In SGD-based MCL, however, the lengthening of the training stream exposes models to more SGD updates, which can exacerbate catastrophic forgetting of previous tasks. On the other hand, the SB-MCL family demonstrates a remarkable level of robustness in many-shot settings. As the number of shots increases, their performance even shows a slight improvement. This observation aligns

Table 2: 10-task 10-shot MCL experiments. We report classification errors for the classification benchmarks while reporting losses for others.

| Method | Classification | | | Regression | | Generation | |
| | Omniglot | CASIA | Sine | Completion | Rotation | VAE | DDPM |
| --- | --- | --- | --- | --- | --- | --- | --- |
| Standard | $.225^{\pm.075}$ | $.340^{\pm.060}$ | $.012^{\pm.001}$ | $.147^{\pm.013}$ | $.571^{\pm.063}$ | $.675^{\pm.029}$ | $5.070^{\pm.243}$ |
| Online | $.850^{\pm.052}$ | $.950^{\pm.032}$ | $.495^{\pm.053}$ | $.333^{\pm.038}$ | $1.187^{\pm.110}$ | $.851^{\pm.013}$ | $13.797^{\pm.349}$ |
| OML-Reptile | $.136^{\pm.005}$ | $.057^{\pm.003}$ | $.027^{\pm.001}$ | $.104^{\pm.000}$ | $.050^{\pm.002}$ | $.454^{\pm.000}$ | $3.531^{\pm.010}$ |
| OML | $.046^{\pm.002}$ | $.015^{\pm.001}$ | $.016^{\pm.001}$ | $.105^{\pm.000}$ | $.053^{\pm.002}$ | $.442^{\pm.003}$ | $3.530^{\pm.018}$ |
| SB-MCL | $\mathbf{.008}^{\pm.000}$ | $\mathbf{.002}^{\pm.000}$ | $\mathbf{.001}^{\pm.000}$ | $\mathbf{.100}^{\pm.001}$ | $\mathbf{.039}^{\pm.001}$ | $\mathbf{.428}^{\pm.001}$ | $\mathbf{3.448}^{\pm.010}$ |

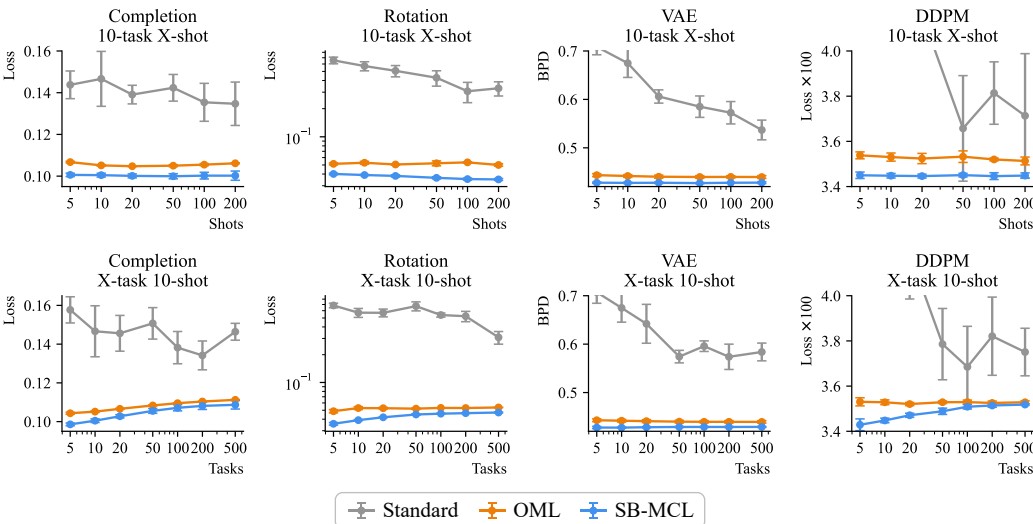

Figure 3: Extrapolation experiments of image completion, rotation prediction, and deep generative modeling. Each method is meta-trained with 10-task 10-shot episodes and tested on meta-test episodes with different length configurations. All scores are better when lower. Best viewed in color.

with our formulation. Since our posterior follows an exponential family distribution with sufficient statistics of fixed size, maintaining a fixed number of tasks while increasing the number of shots only serves to enhance the accuracy of the variational posterior.

**Forgetting vs. Underfitting.** Although SB-MCL is robust to many-shot settings, its performance degrades as it encounters more tasks, which can be observed in the X-task 10-shot plots. However, since SB-MCL will yield the same results even if all the tasks are provided at once (multi-task learning), the degradation should be considered underfitting rather than forgetting. Therefore, increasing the CL performance in SB-MCL becomes an architectural problem. In other words, improving the representational capacity of the overall architecture (including the model, the learner, and the variational posterior) entails improved CL performance. This is a crucial difference from the SGD-based MCL, where the CL performance is not necessarily aligned well with the model capacity.

## 6 CONCLUSION

This work introduces a general MCL framework that combines simple statistical models' robustness to forgetting and the flexibility of neural networks. Its superior performance is also empirically demonstrated in diverse domains. Unifying several prior works under the same framework, we aim to establish a solid foundation for the future sequential Bayesian approaches in the field of MCL. As discussed in §5.3, our framework transforms CL's forgetting issue into an underfitting problem. This allows us to approach MCL as an architectural problem, designing neural architectures that can effectively interact with statistical models, which can be an exciting avenue for further research.

REPRODUCIBILITY STATEMENT

We are fully committed to ensuring the complete reproducibility of our research. We have included the PyTorch (Paszke et al., 2019) code and specific commands in the supplementary material, allowing anyone to easily replicate every single experiment, including the baselines. All the datasets used in our experiments are publicly available on the web, and our code is designed to automatically download the necessary datasets before starting the experiments. We believe that our code will serve as a valuable resource for the community, particularly for newcomers in the field of MCL, providing them with a solid foundation for their research endeavors.

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

## A   VARIATIONAL BOUND DERIVATION

The derivation of the variational bound for supervised learning setup (Eq. 1) is as follows:

$$
\begin{aligned}
&\log p_\theta(\tilde{y}_{1:N}|\tilde{x}_{1:N}, \mathcal{D}) \\
&= -\log p_\theta(z|\tilde{y}_{1:N}, \tilde{x}_{1:N}, \mathcal{D}) + \log p_\theta(\tilde{y}_{1:N}, z|\tilde{x}_{1:N}, \mathcal{D}) \\
&= \mathbb{E}_{z \sim q_\phi(z|\mathcal{D})} \left[ \log q_\phi(z|\mathcal{D}) - \log p_\theta(z|\tilde{y}_{1:N}, \tilde{x}_{1:N}, \mathcal{D}) + \log p_\theta(\tilde{y}_{1:N}, z|\tilde{x}_{1:N}, \mathcal{D}) - \log q_\phi(z|\mathcal{D}) \right] \\
&= D_{\mathrm{KL}}\left( q_\phi(z|\mathcal{D}) \,\|\, p_\theta(z|\tilde{y}_{1:N}, \tilde{x}_{1:N}, \mathcal{D}) \right) + \mathbb{E}_{z \sim q_\phi(z|\mathcal{D})} \left[ \log p_\theta(\tilde{y}_{1:N}, z|\tilde{x}_{1:N}, \mathcal{D}) - \log q_\phi(z|\mathcal{D}) \right] \\
&\geq \mathbb{E}_{z \sim q_\phi(z|\mathcal{D})} \left[ \log p_\theta(\tilde{y}_{1:N}, z|\tilde{x}_{1:N}, \mathcal{D}) - \log q_\phi(z|\mathcal{D}) \right] \\
&= \mathbb{E}_{z \sim q_\phi(z|\mathcal{D})} \left[ \log p_\theta(\tilde{y}_{1:N}|z, \tilde{x}_{1:N}) + \log p_\theta(z|\tilde{x}_{1:N}, \mathcal{D}) - \log q_\phi(z|\mathcal{D}) \right] \qquad (8) \\
&= \mathbb{E}_{z \sim q_\phi(z|\mathcal{D})} \big[ \log p_\theta(\tilde{y}_{1:N}|z, \tilde{x}_{1:N}) + \log p_\theta(\mathcal{D}|z, \tilde{x}_{1:N}) + \log p_\theta(z|\tilde{x}_{1:N}) - \log p_\theta(\mathcal{D}|\tilde{x}_{1:N}) \\
&\qquad\qquad\qquad - \log q_\phi(z|\mathcal{D}) \big] \\
&= \mathbb{E}_{z \sim q_\phi(z|\mathcal{D})} \left[ \log p_\theta(\tilde{y}_{1:N}|z, \tilde{x}_{1:N}) + \log p_\theta(\mathcal{D}|z) + \log p_\theta(z) - \log p_\theta(\mathcal{D}) - \log q_\phi(z|\mathcal{D}) \right] \\
&= \mathbb{E}_{z \sim q_\phi(z|\mathcal{D})} \left[ \log p_\theta(\tilde{y}_{1:N}|z, \tilde{x}_{1:N}) + \log p_\theta(\mathcal{D}|z) \right] - D_{\mathrm{KL}}\left( q_\phi(z|\mathcal{D}) \,\|\, p_\theta(z) \right) - \log p_\theta(\mathcal{D}) \\
&= \mathbb{E}_{z \sim q_\phi(z|\mathcal{D})} \left[ \sum_{n=1}^{N} \log p_\theta(\tilde{y}_n|\tilde{x}_n, z) + \sum_{t=1}^{T} \log p_\theta(y_t|x_t, z) \right] \\
&\qquad - D_{\mathrm{KL}}\left( q_\phi(z|\mathcal{D}) \,\|\, p_\theta(z) \right) - \underbrace{\log p_\theta(\mathcal{D})}_{\mathrm{const.}}
\end{aligned}
$$

We can derive a similar bound for unsupervised settings (Eq. 2):

$$
\begin{aligned}
&\log p_\theta(\tilde{x}_{1:N}|\mathcal{D}) \\
&= -\log p_\theta(z|\tilde{x}_{1:N}, \mathcal{D}) + \log p_\theta(\tilde{x}_{1:N}, z|\mathcal{D}) \\
&= \mathbb{E}_{z \sim q_\phi(z|\mathcal{D})} \left[ \log q_\phi(z|\mathcal{D}) - \log p_\theta(z|\tilde{x}_{1:N}, \mathcal{D}) + \log p_\theta(\tilde{x}_{1:N}, z|\mathcal{D}) - \log q_\phi(z|\mathcal{D}) \right] \\
&= D_{\mathrm{KL}}\left( q_\phi(z|\mathcal{D}) \,\|\, p_\theta(z|\mathcal{D}) \right) + \mathbb{E}_{z \sim q_\phi(z|\mathcal{D})} \left[ \log p_\theta(\tilde{x}_{1:N}, z|\mathcal{D}) - \log q_\phi(z|\mathcal{D}) \right] \\
&\geq \mathbb{E}_{z \sim q_\phi(z|\mathcal{D})} \left[ \log p_\theta(\tilde{x}_{1:N}, z|\mathcal{D}) - \log q_\phi(z|\mathcal{D}) \right] \\
&= \mathbb{E}_{z \sim q_\phi(z|\mathcal{D})} \left[ \log p_\theta(\tilde{x}_{1:N}|z, \mathcal{D}) + \log p_\theta(z|\mathcal{D}) - \log q_\phi(z|\mathcal{D}) \right] \\
&= \mathbb{E}_{z \sim q_\phi(z|\mathcal{D})} \left[ \log p_\theta(\tilde{x}_{1:N}|z) + \log p_\theta(\mathcal{D}|z) + \log p_\theta(z) - \log p_\theta(\mathcal{D}) - \log q_\phi(z|\mathcal{D}) \right] \\
&= \mathbb{E}_{z \sim q_\phi(z|\mathcal{D})} \left[ \log p_\theta(\tilde{x}_{1:N}|z) + \log p_\theta(\mathcal{D}|z) \right] - D_{\mathrm{KL}}\left( q_\phi(z|\mathcal{D}) \,\|\, p_\theta(z) \right) - \log p_\theta(\mathcal{D}) \\
&= \mathbb{E}_{z \sim q_\phi(z|\mathcal{D})} \left[ \sum_{n=1}^{N} \log p_\theta(\tilde{x}_n|z) + \sum_{t=1}^{T} \log p_\theta(x_t|z) \right] - D_{\mathrm{KL}}\left( q_\phi(z|\mathcal{D}) \,\|\, p_\theta(z) \right) - \underbrace{\log p_\theta(\mathcal{D})}_{\mathrm{const.}}
\end{aligned}
$$

It is noteworthy that Neural Process (Garnelo et al., 2018b) instead approximates $\log p_\theta(z|\tilde{x}_{1:N}, \mathcal{D})$ of Eq. 8 with $\log q_\phi(z|\tilde{x}_{1:N}, \mathcal{D})$:

$$
\begin{aligned}
&\mathbb{E}_{z \sim q_\phi(z|\mathcal{D})} \left[ \log p_\theta(\tilde{y}_{1:N}|z, \tilde{x}_{1:N}) + \log p_\theta(z|\tilde{x}_{1:N}, \mathcal{D}) - \log q_\phi(z|\mathcal{D}) \right] \\
&\approx \mathbb{E}_{z \sim q_\phi(z|\mathcal{D})} \left[ \log p_\theta(\tilde{y}_{1:N}|z, \tilde{x}_{1:N}) + \log q_\phi(z|\tilde{x}_{1:N}, \mathcal{D}) - \log q_\phi(z|\mathcal{D}) \right] \\
&= \mathbb{E}_{z \sim q_\phi(z|\mathcal{D})} \left[ \sum_{n=1}^{N} \log p_\theta(\tilde{y}_n|\tilde{x}_n, z) \right] - D_{\mathrm{KL}}\left( q_\phi(z|\mathcal{D}) \| q_\phi(z|\tilde{x}_{1:N}, \mathcal{D}) \right)
\end{aligned}
$$

Since we can use the Bayes rule to convert $\log p_\theta(z|\tilde{x}_{1:N}, \mathcal{D})$ into $\log p_\theta(\mathcal{D}|z, \tilde{x}_{1:N}) + \log p_\theta(z|\tilde{x}_{1:N}) - \log p_\theta(\mathcal{D}|\tilde{x}_{1:N})$, which is subsequently reduced to $\log p_\theta(\mathcal{D}|z) + \log p_\theta(z) - \log p_\theta(\mathcal{D})$ by conditional independence, such an approximation is not necessary.

## B   QUALITATIVE EXAMPLES OF DEEP GENERATIVE MCL

In Fig. 4-7, we present qualitative examples of the deep generative model experiments from Fig. 3. For each meta-trained MCL method, we train a VAE and a DDPM on the 5-task 10-shot training

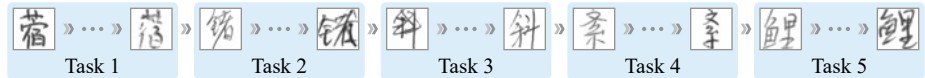

Figure 4: Training stream.

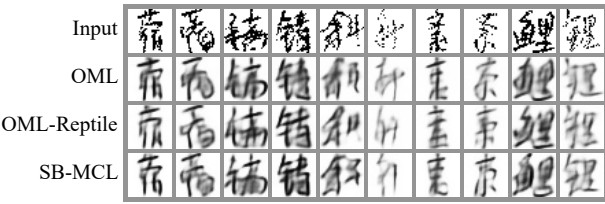

Figure 5: VAE reconstruction samples.

stream in Fig. 4, which is sampled from the meta-test set. Then, we extract 20 generation samples for the VAE (Fig. 6) and the DDPM (Fig. 7). For the VAE, we also visualize the reconstructions of the test images in Fig. 5.

Although the scores of OML and OML-Reptile are much worse than SB-MCL in Fig. 3, the reconstruction results in Fig. 5 do not show a much difference except that the SB-MCL produces slightly better reconstructions. However, the generation results of OML and OML-Reptile are not properly structured, showing that OML and OML-Reptile have difficulty in training VAE on a non-stationary stream. On the other hand, the VAE with SB-MCL produces significantly better samples, demonstrating the effectiveness of our approach.

All the DDPM samples in Fig. 7 are of much higher quality compared to VAE and are hard to distinguish from real images. Since the DDPMs meta-learn general concepts of CASIA images from the large-scale meta-training set, they can produce high-fidelity images. The key difference to notice is whether the DDPM has learned new knowledge from the training stream. Since the training stream is from the meta-test set, it cannot produce the classes in the training stream unless it actually learns from it. Among the samples from OML and OML-Reptile, it is hard to find the classes in the training stream, suggesting that they are producing samples from the meta-training distribution. On the other hand, the DDPM with SB-MCL produces samples remarkably similar to the ones in Fig. 4. This experiment confirms that SB-MCL can be an effective solution for modern deep generative models.

## C  EXTENDED EXPERIMENTAL RESULTS

Table 3: CASIA Classification 10-task X-shot (Error)

| Shots | 5 | 10 | 20 | 50 | 100 | 200 |
|---|---|---|---|---|---|---|
| Standard | $0.540^{\pm0.057}$ | $0.340^{\pm0.060}$ | $0.335^{\pm0.035}$ | $0.090^{\pm0.033}$ | $0.037^{\pm0.016}$ | $0.040^{\pm0.018}$ |
| Online | $0.875^{\pm0.073}$ | $0.950^{\pm0.032}$ | $0.825^{\pm0.051}$ | $0.850^{\pm0.052}$ | $0.800^{\pm0.092}$ | $0.550^{\pm0.116}$ |
| OML-Reptile | $0.049^{\pm0.002}$ | $0.057^{\pm0.003}$ | $0.066^{\pm0.002}$ | $0.083^{\pm0.004}$ | $0.101^{\pm0.002}$ | $0.121^{\pm0.003}$ |
| OML | $0.012^{\pm0.001}$ | $0.015^{\pm0.001}$ | $0.019^{\pm0.001}$ | $0.026^{\pm0.002}$ | $0.031^{\pm0.002}$ | $0.039^{\pm0.001}$ |
| PN | $0.003^{\pm0.000}$ | $0.002^{\pm0.000}$ | $0.002^{\pm0.000}$ | $0.001^{\pm0.000}$ | $0.001^{\pm0.000}$ | $0.001^{\pm0.000}$ |
| GeMCL-MAP | $0.003^{\pm0.000}$ | $0.002^{\pm0.000}$ | $0.002^{\pm0.000}$ | $0.001^{\pm0.000}$ | $0.001^{\pm0.000}$ | $0.001^{\pm0.000}$ |
| GeMCL | $0.003^{\pm0.000}$ | $0.002^{\pm0.000}$ | $0.002^{\pm0.000}$ | $0.001^{\pm0.000}$ | $0.001^{\pm0.000}$ | $0.002^{\pm0.000}$ |

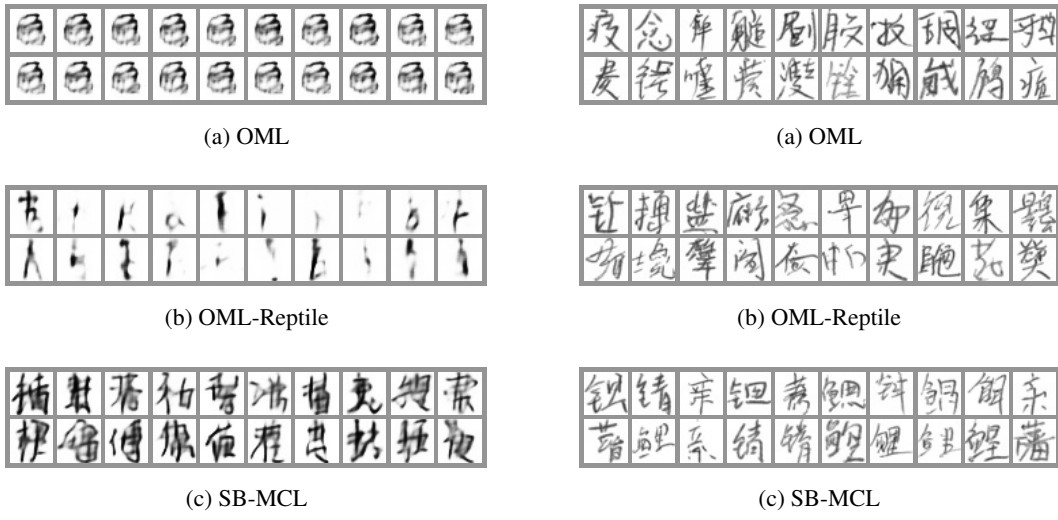

(a) OML

(a) OML

(b) OML-Reptile

(b) OML-Reptile

(c) SB-MCL

(c) SB-MCL

Figure 6: VAE generation samples.

Figure 7: DDPM generation samples.

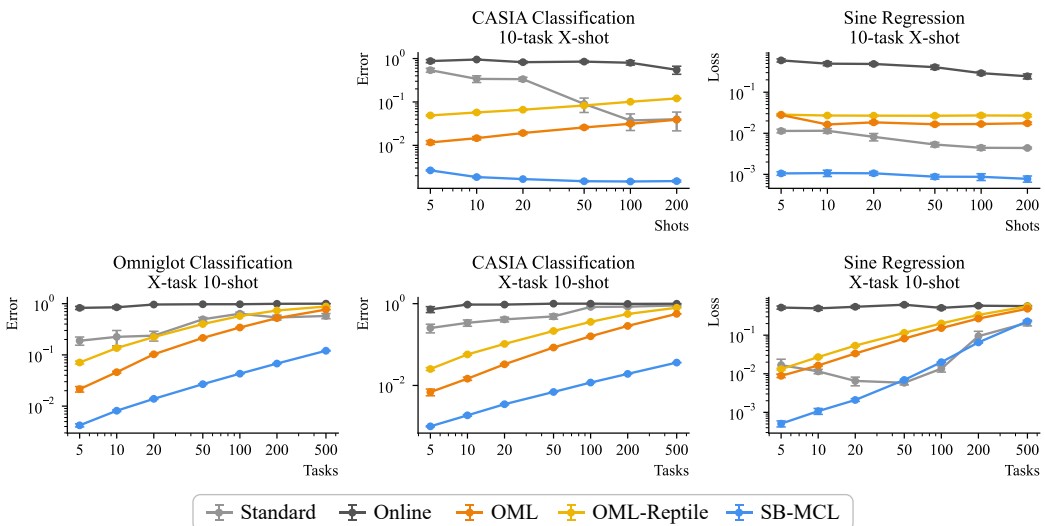

Figure 8: Extrapolation experiments of image classification and sine wave regression. Each method is meta-trained with 10-task 10-shot episodes and tested on meta-test episodes with different length configurations. For classification, the scores of GeMCL are reported as SB-MCL, while ALPaCA's score is reported as SB-MCL for the sine wave regression. All scores are better when lower. Best viewed in color.

Table 4: Sine Regression 10-task X-shot (Loss)

| Shots | 5 | 10 | 20 | 50 | 100 | 200 |
|---|---|---|---|---|---|---|
| Standard | $0.011^{\pm 0.001}$ | $0.012^{\pm 0.001}$ | $0.008^{\pm 0.002}$ | $0.005^{\pm 0.001}$ | $0.004^{\pm 0.001}$ | $0.004^{\pm 0.000}$ |
| Online | $0.595^{\pm 0.052}$ | $0.495^{\pm 0.053}$ | $0.487^{\pm 0.038}$ | $0.407^{\pm 0.047}$ | $0.291^{\pm 0.028}$ | $0.244^{\pm 0.034}$ |
| OML-Reptile | $0.028^{\pm 0.002}$ | $0.027^{\pm 0.001}$ | $0.027^{\pm 0.001}$ | $0.027^{\pm 0.002}$ | $0.027^{\pm 0.002}$ | $0.027^{\pm 0.002}$ |
| OML | $0.028^{\pm 0.001}$ | $0.016^{\pm 0.001}$ | $0.018^{\pm 0.001}$ | $0.017^{\pm 0.001}$ | $0.017^{\pm 0.001}$ | $0.018^{\pm 0.001}$ |
| ALPaCA | $0.001^{\pm 0.000}$ | $0.001^{\pm 0.000}$ | $0.001^{\pm 0.000}$ | $0.001^{\pm 0.000}$ | $0.001^{\pm 0.000}$ | $0.001^{\pm 0.000}$ |

Table 5: Omniglot Classification X-task 10-shot (Error)

| Tasks | 5 | 10 | 20 | 50 | 100 | 200 | 500 |
|---|---|---|---|---|---|---|---|
| Standard | $0.189^{\pm0.034}$ | $0.225^{\pm0.075}$ | $0.237^{\pm0.051}$ | $0.495^{\pm0.043}$ | $0.637^{\pm0.045}$ | $0.537^{\pm0.059}$ | $0.575^{\pm0.067}$ |
| Online | $0.828^{\pm0.057}$ | $0.850^{\pm0.052}$ | $0.963^{\pm0.018}$ | $0.975^{\pm0.015}$ | $0.975^{\pm0.016}$ | $0.994^{\pm0.006}$ | $1.000^{\pm0.000}$ |
| OML-Reptile | $0.071^{\pm0.005}$ | $0.136^{\pm0.005}$ | $0.223^{\pm0.006}$ | $0.400^{\pm0.006}$ | $0.573^{\pm0.005}$ | $0.736^{\pm0.003}$ | $0.880^{\pm0.002}$ |
| OML | $0.021^{\pm0.003}$ | $0.046^{\pm0.002}$ | $0.103^{\pm0.002}$ | $0.215^{\pm0.004}$ | $0.343^{\pm0.005}$ | $0.522^{\pm0.004}$ | $0.767^{\pm0.002}$ |
| PN | $0.004^{\pm0.001}$ | $0.008^{\pm0.000}$ | $0.014^{\pm0.000}$ | $0.026^{\pm0.001}$ | $0.042^{\pm0.000}$ | $0.065^{\pm0.000}$ | $0.117^{\pm0.000}$ |
| GeMCL-MAP | $0.005^{\pm0.000}$ | $0.008^{\pm0.001}$ | $0.014^{\pm0.000}$ | $0.027^{\pm0.001}$ | $0.043^{\pm0.001}$ | $0.068^{\pm0.001}$ | $0.121^{\pm0.002}$ |
| GeMCL | $0.004^{\pm0.000}$ | $0.008^{\pm0.000}$ | $0.014^{\pm0.000}$ | $0.027^{\pm0.000}$ | $0.043^{\pm0.000}$ | $0.068^{\pm0.001}$ | $0.120^{\pm0.002}$ |

Table 6: CASIA Classification X-task 10-shot (Error)

| Tasks | 5 | 10 | 20 | 50 | 100 | 200 | 500 |
|---|---|---|---|---|---|---|---|
| Standard | $0.254^{\pm0.061}$ | $0.340^{\pm0.060}$ | $0.412^{\pm0.055}$ | $0.487^{\pm0.069}$ | $0.831^{\pm0.048}$ | $0.844^{\pm0.041}$ | $0.906^{\pm0.033}$ |
| Online | $0.725^{\pm0.120}$ | $0.950^{\pm0.032}$ | $0.950^{\pm0.032}$ | $1.000^{\pm0.000}$ | $1.000^{\pm0.000}$ | $0.988^{\pm0.012}$ | $0.995^{\pm0.005}$ |
| OML-Reptile | $0.025^{\pm0.002}$ | $0.057^{\pm0.003}$ | $0.104^{\pm0.002}$ | $0.215^{\pm0.004}$ | $0.359^{\pm0.002}$ | $0.559^{\pm0.005}$ | $0.796^{\pm0.003}$ |
| OML | $0.007^{\pm0.001}$ | $0.015^{\pm0.001}$ | $0.033^{\pm0.001}$ | $0.085^{\pm0.001}$ | $0.159^{\pm0.001}$ | $0.286^{\pm0.002}$ | $0.564^{\pm0.001}$ |
| PN | $0.001^{\pm0.000}$ | $0.002^{\pm0.000}$ | $0.003^{\pm0.000}$ | $0.007^{\pm0.000}$ | $0.012^{\pm0.000}$ | $0.019^{\pm0.000}$ | $0.036^{\pm0.001}$ |
| GeMCL-MAP | $0.001^{\pm0.000}$ | $0.002^{\pm0.000}$ | $0.003^{\pm0.000}$ | $0.007^{\pm0.000}$ | $0.012^{\pm0.000}$ | $0.019^{\pm0.000}$ | $0.036^{\pm0.000}$ |
| GeMCL | $0.001^{\pm0.000}$ | $0.002^{\pm0.000}$ | $0.003^{\pm0.000}$ | $0.007^{\pm0.000}$ | $0.012^{\pm0.000}$ | $0.019^{\pm0.000}$ | $0.036^{\pm0.000}$ |

Table 7: Sine Regression X-task 10-shot (Error)

| Tasks | 5 | 10 | 20 | 50 | 100 | 200 | 500 |
|---|---|---|---|---|---|---|---|
| Standard | $0.017^{\pm0.007}$ | $0.012^{\pm0.001}$ | $0.007^{\pm0.002}$ | $0.006^{\pm0.001}$ | $0.014^{\pm0.003}$ | $0.094^{\pm0.032}$ | $0.210^{\pm0.036}$ |
| Online | $0.520^{\pm0.049}$ | $0.495^{\pm0.053}$ | $0.544^{\pm0.046}$ | $0.615^{\pm0.040}$ | $0.510^{\pm0.044}$ | $0.580^{\pm0.030}$ | $0.567^{\pm0.020}$ |
| OML-Reptile | $0.013^{\pm0.001}$ | $0.027^{\pm0.001}$ | $0.054^{\pm0.002}$ | $0.115^{\pm0.003}$ | $0.201^{\pm0.004}$ | $0.335^{\pm0.005}$ | $0.559^{\pm0.003}$ |
| OML | $0.009^{\pm0.001}$ | $0.016^{\pm0.001}$ | $0.034^{\pm0.002}$ | $0.082^{\pm0.001}$ | $0.153^{\pm0.000}$ | $0.270^{\pm0.001}$ | $0.484^{\pm0.002}$ |
| ALPaCA | $0.001^{\pm0.000}$ | $0.001^{\pm0.000}$ | $0.002^{\pm0.000}$ | $0.007^{\pm0.000}$ | $0.020^{\pm0.000}$ | $0.065^{\pm0.001}$ | $0.228^{\pm0.001}$ |

Table 8: Completion 10-task X-shot (Loss)

| Shots | 5 | 10 | 20 | 50 | 100 | 200 |
|---|---|---|---|---|---|---|
| Standard | $0.144^{\pm0.007}$ | $0.147^{\pm0.013}$ | $0.139^{\pm0.004}$ | $0.142^{\pm0.006}$ | $0.135^{\pm0.009}$ | $0.135^{\pm0.010}$ |
| Online | $0.499^{\pm0.032}$ | $0.333^{\pm0.038}$ | $0.204^{\pm0.012}$ | $0.155^{\pm0.009}$ | $0.155^{\pm0.013}$ | $0.175^{\pm0.011}$ |
| Reptile | $0.125^{\pm0.001}$ | $0.126^{\pm0.001}$ | $0.127^{\pm0.000}$ | $0.129^{\pm0.000}$ | $0.130^{\pm0.000}$ | $0.132^{\pm0.001}$ |
| MAML | $0.125^{\pm0.003}$ | $0.108^{\pm0.000}$ | $0.108^{\pm0.001}$ | $0.108^{\pm0.000}$ | $0.109^{\pm0.000}$ | $0.110^{\pm0.000}$ |
| OML-Reptile | $0.105^{\pm0.000}$ | $0.104^{\pm0.000}$ | $0.104^{\pm0.000}$ | $0.105^{\pm0.000}$ | $0.106^{\pm0.000}$ | $0.107^{\pm0.000}$ |
| OML | $0.107^{\pm0.000}$ | $0.105^{\pm0.000}$ | $0.105^{\pm0.000}$ | $0.105^{\pm0.000}$ | $0.106^{\pm0.000}$ | $0.106^{\pm0.000}$ |
| SB-MCL (MAP) | $0.101^{\pm0.001}$ | $0.100^{\pm0.001}$ | $0.100^{\pm0.001}$ | $0.100^{\pm0.001}$ | $0.100^{\pm0.002}$ | $0.100^{\pm0.002}$ |
| SB-MCL | $0.101^{\pm0.001}$ | $0.100^{\pm0.001}$ | $0.100^{\pm0.001}$ | $0.100^{\pm0.001}$ | $0.100^{\pm0.002}$ | $0.100^{\pm0.002}$ |

Table 9: Rotation 10-task X-shot (Loss)

| Shots | 5 | 10 | 20 | 50 | 100 | 200 |
|---|---|---|---|---|---|---|
| Standard | $0.657^{\pm0.054}$ | $0.571^{\pm0.063}$ | $0.507^{\pm0.070}$ | $0.428^{\pm0.080}$ | $0.307^{\pm0.075}$ | $0.330^{\pm0.057}$ |
| Online | $0.957^{\pm0.101}$ | $1.187^{\pm0.110}$ | $0.775^{\pm0.062}$ | $0.932^{\pm0.115}$ | $0.765^{\pm0.084}$ | $0.779^{\pm0.126}$ |
| Reptile | $0.136^{\pm0.010}$ | $0.171^{\pm0.006}$ | $0.209^{\pm0.010}$ | $0.317^{\pm0.011}$ | $0.401^{\pm0.008}$ | $0.472^{\pm0.014}$ |
| MAML | $0.968^{\pm0.024}$ | $0.965^{\pm0.024}$ | $0.959^{\pm0.024}$ | $0.964^{\pm0.013}$ | $0.969^{\pm0.010}$ | $0.959^{\pm0.012}$ |
| OML-Reptile | $0.050^{\pm0.002}$ | $0.050^{\pm0.002}$ | $0.050^{\pm0.001}$ | $0.047^{\pm0.001}$ | $0.046^{\pm0.001}$ | $0.045^{\pm0.000}$ |
| OML | $0.051^{\pm0.001}$ | $0.053^{\pm0.002}$ | $0.051^{\pm0.001}$ | $0.052^{\pm0.003}$ | $0.053^{\pm0.001}$ | $0.050^{\pm0.001}$ |
| SB-MCL (MAP) | $0.040^{\pm0.001}$ | $0.040^{\pm0.001}$ | $0.039^{\pm0.001}$ | $0.036^{\pm0.001}$ | $0.036^{\pm0.001}$ | $0.035^{\pm0.001}$ |
| SB-MCL | $0.040^{\pm0.001}$ | $0.039^{\pm0.001}$ | $0.038^{\pm0.001}$ | $0.036^{\pm0.001}$ | $0.035^{\pm0.001}$ | $0.035^{\pm0.001}$ |

Table 10: VAE 10-task X-shot (BPD)

| Shots | 5 | 10 | 20 | 50 | 100 | 200 |
|---|---|---|---|---|---|---|
| Standard | $0.709^{\pm0.017}$ | $0.675^{\pm0.029}$ | $0.606^{\pm0.014}$ | $0.585^{\pm0.022}$ | $0.572^{\pm0.023}$ | $0.537^{\pm0.020}$ |
| Online | $0.921^{\pm0.010}$ | $0.851^{\pm0.013}$ | $0.787^{\pm0.016}$ | $0.743^{\pm0.035}$ | $0.748^{\pm0.023}$ | $0.713^{\pm0.022}$ |
| Reptile | $0.768^{\pm0.000}$ | $0.766^{\pm0.001}$ | $0.766^{\pm0.001}$ | $0.768^{\pm0.001}$ | $0.767^{\pm0.001}$ | $0.768^{\pm0.000}$ |
| OML-Reptile | $0.454^{\pm0.002}$ | $0.454^{\pm0.000}$ | $0.455^{\pm0.002}$ | $0.455^{\pm0.002}$ | $0.456^{\pm0.001}$ | $0.459^{\pm0.001}$ |
| OML | $0.444^{\pm0.003}$ | $0.442^{\pm0.003}$ | $0.440^{\pm0.003}$ | $0.440^{\pm0.003}$ | $0.440^{\pm0.002}$ | $0.440^{\pm0.003}$ |
| SB-MCL (MAP) | $0.428^{\pm0.001}$ | $0.428^{\pm0.001}$ | $0.427^{\pm0.001}$ | $0.428^{\pm0.001}$ | $0.428^{\pm0.001}$ | $0.428^{\pm0.001}$ |
| SB-MCL | $0.428^{\pm0.001}$ | $0.428^{\pm0.001}$ | $0.428^{\pm0.001}$ | $0.427^{\pm0.001}$ | $0.428^{\pm0.000}$ | $0.428^{\pm0.002}$ |

Table 11: DDPM 10-task X-shot (Loss ×100)

| Shots | 5 | 10 | 20 | 50 | 100 | 200 |
|---|---|---|---|---|---|---|
| Standard | $4.927^{\pm 0.254}$ | $5.070^{\pm 0.243}$ | $4.162^{\pm 0.138}$ | $3.657^{\pm 0.234}$ | $3.814^{\pm 0.138}$ | $3.713^{\pm 0.275}$ |
| Online | $17.132^{\pm 0.324}$ | $13.797^{\pm 0.349}$ | $10.797^{\pm 0.391}$ | $8.034^{\pm 0.347}$ | $7.130^{\pm 0.228}$ | $6.128^{\pm 0.260}$ |
| OML-Reptile | $3.529^{\pm 0.009}$ | $3.531^{\pm 0.010}$ | $3.530^{\pm 0.022}$ | $3.522^{\pm 0.011}$ | $3.527^{\pm 0.022}$ | $3.516^{\pm 0.009}$ |
| OML | $3.538^{\pm 0.015}$ | $3.530^{\pm 0.018}$ | $3.524^{\pm 0.023}$ | $3.532^{\pm 0.026}$ | $3.520^{\pm 0.006}$ | $3.513^{\pm 0.017}$ |
| SB-MCL (MAP) | $3.454^{\pm 0.011}$ | $3.454^{\pm 0.010}$ | $3.449^{\pm 0.009}$ | $3.447^{\pm 0.004}$ | $3.443^{\pm 0.011}$ | $3.459^{\pm 0.012}$ |
| SB-MCL | $3.450^{\pm 0.014}$ | $3.448^{\pm 0.010}$ | $3.446^{\pm 0.009}$ | $3.451^{\pm 0.008}$ | $3.446^{\pm 0.015}$ | $3.449^{\pm 0.012}$ |

Table 12: Completion X-task 10-shot (Loss)

| Tasks | 5 | 10 | 20 | 50 | 100 | 200 | 500 |
|---|---|---|---|---|---|---|---|
| Standard | $0.158^{\pm 0.007}$ | $0.147^{\pm 0.013}$ | $0.146^{\pm 0.009}$ | $0.151^{\pm 0.008}$ | $0.138^{\pm 0.008}$ | $0.134^{\pm 0.007}$ | $0.146^{\pm 0.004}$ |
| Online | $0.504^{\pm 0.034}$ | $0.333^{\pm 0.038}$ | $0.179^{\pm 0.010}$ | $0.169^{\pm 0.008}$ | $0.148^{\pm 0.012}$ | $0.173^{\pm 0.006}$ | $0.158^{\pm 0.006}$ |
| Reptile | $0.124^{\pm 0.000}$ | $0.126^{\pm 0.000}$ | $0.127^{\pm 0.000}$ | $0.128^{\pm 0.000}$ | $0.128^{\pm 0.000}$ | $0.129^{\pm 0.000}$ | $0.129^{\pm 0.000}$ |
| MAML | $0.123^{\pm 0.002}$ | $0.108^{\pm 0.000}$ | $0.108^{\pm 0.000}$ | $0.110^{\pm 0.000}$ | $0.110^{\pm 0.000}$ | $0.110^{\pm 0.000}$ | $0.111^{\pm 0.000}$ |
| OML-Reptile | $0.104^{\pm 0.000}$ | $0.104^{\pm 0.000}$ | $0.106^{\pm 0.000}$ | $0.107^{\pm 0.000}$ | $0.108^{\pm 0.000}$ | $0.108^{\pm 0.000}$ | $0.109^{\pm 0.000}$ |
| OML | $0.104^{\pm 0.001}$ | $0.105^{\pm 0.000}$ | $0.107^{\pm 0.000}$ | $0.108^{\pm 0.000}$ | $0.110^{\pm 0.000}$ | $0.110^{\pm 0.000}$ | $0.111^{\pm 0.000}$ |
| SB-MCL (MAP) | $0.099^{\pm 0.001}$ | $0.100^{\pm 0.001}$ | $0.103^{\pm 0.001}$ | $0.106^{\pm 0.001}$ | $0.107^{\pm 0.002}$ | $0.108^{\pm 0.002}$ | $0.109^{\pm 0.002}$ |
| SB-MCL | $0.099^{\pm 0.001}$ | $0.100^{\pm 0.001}$ | $0.103^{\pm 0.001}$ | $0.106^{\pm 0.001}$ | $0.107^{\pm 0.002}$ | $0.108^{\pm 0.002}$ | $0.109^{\pm 0.002}$ |

Table 13: Rotation X-task 10-shot (Loss)

| Tasks | 5 | 10 | 20 | 50 | 100 | 200 | 500 |
|---|---|---|---|---|---|---|---|
| Standard | $0.684^{\pm 0.039}$ | $0.571^{\pm 0.063}$ | $0.569^{\pm 0.055}$ | $0.672^{\pm 0.075}$ | $0.538^{\pm 0.028}$ | $0.523^{\pm 0.067}$ | $0.308^{\pm 0.048}$ |
| Online | $0.959^{\pm 0.073}$ | $1.187^{\pm 0.110}$ | $1.023^{\pm 0.101}$ | $1.006^{\pm 0.128}$ | $0.973^{\pm 0.090}$ | $0.917^{\pm 0.071}$ | $1.024^{\pm 0.043}$ |
| Reptile | $0.135^{\pm 0.002}$ | $0.171^{\pm 0.006}$ | $0.221^{\pm 0.011}$ | $0.341^{\pm 0.015}$ | $0.467^{\pm 0.013}$ | $0.557^{\pm 0.010}$ | $0.609^{\pm 0.017}$ |
| MAML | $0.962^{\pm 0.025}$ | $0.965^{\pm 0.024}$ | $0.964^{\pm 0.023}$ | $0.973^{\pm 0.013}$ | $0.976^{\pm 0.009}$ | $0.978^{\pm 0.008}$ | $0.977^{\pm 0.007}$ |
| OML-Reptile | $0.048^{\pm 0.003}$ | $0.050^{\pm 0.002}$ | $0.050^{\pm 0.001}$ | $0.052^{\pm 0.001}$ | $0.053^{\pm 0.001}$ | $0.055^{\pm 0.001}$ | $0.056^{\pm 0.001}$ |
| OML | $0.049^{\pm 0.002}$ | $0.053^{\pm 0.002}$ | $0.053^{\pm 0.001}$ | $0.052^{\pm 0.001}$ | $0.053^{\pm 0.000}$ | $0.053^{\pm 0.000}$ | $0.054^{\pm 0.000}$ |
| SB-MCL (MAP) | $0.035^{\pm 0.002}$ | $0.040^{\pm 0.001}$ | $0.042^{\pm 0.001}$ | $0.045^{\pm 0.001}$ | $0.046^{\pm 0.000}$ | $0.047^{\pm 0.000}$ | $0.047^{\pm 0.000}$ |
| SB-MCL | $0.036^{\pm 0.001}$ | $0.039^{\pm 0.001}$ | $0.042^{\pm 0.000}$ | $0.045^{\pm 0.001}$ | $0.046^{\pm 0.000}$ | $0.047^{\pm 0.000}$ | $0.047^{\pm 0.001}$ |

Table 14: VAE X-task 10-shot (BPD)

| Tasks | 5 | 10 | 20 | 50 | 100 | 200 | 500 |
|---|---|---|---|---|---|---|---|
| Standard | $0.707^{\pm 0.023}$ | $0.675^{\pm 0.029}$ | $0.642^{\pm 0.040}$ | $0.574^{\pm 0.013}$ | $0.596^{\pm 0.011}$ | $0.574^{\pm 0.026}$ | $0.584^{\pm 0.018}$ |
| Online | $0.915^{\pm 0.016}$ | $0.851^{\pm 0.013}$ | $0.781^{\pm 0.018}$ | $0.771^{\pm 0.024}$ | $0.776^{\pm 0.023}$ | $0.748^{\pm 0.011}$ | $0.749^{\pm 0.008}$ |
| Reptile | $0.768^{\pm 0.001}$ | $0.766^{\pm 0.001}$ | $0.766^{\pm 0.001}$ | $0.767^{\pm 0.000}$ | $0.767^{\pm 0.000}$ | $0.767^{\pm 0.000}$ | $0.767^{\pm 0.000}$ |
| OML-Reptile | $0.454^{\pm 0.002}$ | $0.454^{\pm 0.000}$ | $0.455^{\pm 0.001}$ | $0.457^{\pm 0.001}$ | $0.457^{\pm 0.001}$ | $0.458^{\pm 0.001}$ | $0.459^{\pm 0.001}$ |
| OML | $0.443^{\pm 0.003}$ | $0.442^{\pm 0.003}$ | $0.441^{\pm 0.003}$ | $0.440^{\pm 0.003}$ | $0.440^{\pm 0.003}$ | $0.440^{\pm 0.003}$ | $0.439^{\pm 0.003}$ |
| SB-MCL (MAP) | $0.427^{\pm 0.001}$ | $0.428^{\pm 0.001}$ | $0.428^{\pm 0.001}$ | $0.429^{\pm 0.001}$ | $0.429^{\pm 0.001}$ | $0.429^{\pm 0.001}$ | $0.429^{\pm 0.001}$ |
| SB-MCL | $0.428^{\pm 0.002}$ | $0.428^{\pm 0.001}$ | $0.428^{\pm 0.001}$ | $0.429^{\pm 0.001}$ | $0.429^{\pm 0.001}$ | $0.429^{\pm 0.001}$ | $0.429^{\pm 0.001}$ |

Table 15: DDPM X-task 10-shot (Loss ×100)

| Tasks | 5 | 10 | 20 | 50 | 100 | 200 | 500 |
|---|---|---|---|---|---|---|---|
| Standard | $4.989^{\pm 0.280}$ | $5.070^{\pm 0.243}$ | $4.187^{\pm 0.201}$ | $3.786^{\pm 0.158}$ | $3.685^{\pm 0.180}$ | $3.821^{\pm 0.173}$ | $3.710^{\pm 0.109}$ |
| Online | $17.155^{\pm 0.267}$ | $13.797^{\pm 0.349}$ | $10.981^{\pm 0.240}$ | $7.719^{\pm 0.278}$ | $7.034^{\pm 0.266}$ | $6.149^{\pm 0.129}$ | $5.308^{\pm 0.076}$ |
| OML-Reptile | $3.531^{\pm 0.010}$ | $3.531^{\pm 0.010}$ | $3.534^{\pm 0.015}$ | $3.526^{\pm 0.011}$ | $3.525^{\pm 0.007}$ | $3.522^{\pm 0.010}$ | $3.521^{\pm 0.007}$ |
| OML | $3.530^{\pm 0.018}$ | $3.528^{\pm 0.010}$ | $3.521^{\pm 0.007}$ | $3.529^{\pm 0.005}$ | $3.529^{\pm 0.009}$ | $3.525^{\pm 0.008}$ | $3.528^{\pm 0.006}$ |
| SB-MCL (MAP) | $3.450^{\pm 0.022}$ | $3.454^{\pm 0.009}$ | $3.481^{\pm 0.004}$ | $3.496^{\pm 0.007}$ | $3.510^{\pm 0.005}$ | $3.520^{\pm 0.004}$ | $3.525^{\pm 0.006}$ |
| SB-MCL | $3.429^{\pm 0.026}$ | $3.448^{\pm 0.010}$ | $3.471^{\pm 0.009}$ | $3.489^{\pm 0.014}$ | $3.509^{\pm 0.010}$ | $3.514^{\pm 0.008}$ | $3.518^{\pm 0.004}$ |

