# OpenReview forum: "Sequential Bayesian Continual Learning with Meta-Learned Neural Networks"
_ICLR.cc/2024/Conference — Submitted to ICLR 2024_

### Official Review · Reviewer_awoT · 2023-10-31

**Soundness:** 3 good
**Presentation:** 3 good
**Contribution:** 2 fair
**Rating:** 5
**Confidence:** 2

**Summary:**

The paper presents a new general MCL framework, SB-MCL, by considering exponential family posterior distributions for efficient sequential Bayesian updates, and employing a neural network to learn the parameters in the variational prior. The model demonstrates improved performance compared to multiple benchmark datasets.

**Strengths:**

The paper shows extensive empirical comparisons with various benchmark datasets and on multiple tasks, which demonstrates the advantage of the SB-MCL framework.

**Weaknesses:**

The contribution of the paper lies in replacing the inner update rules of MCL with sequential Bayesian updates using exponential family distributions. However, both the idea of Bayesian updates and using exponential family distribution as variational posterior is not new. This paper combines several existing ideas and demonstrates superior performance of the method, but the novelty is somewhat lacking.

**Questions:**

While the paper states a general results for exponential family distributions, the main results are shown with factorized Gaussian. How does this affect the result? Does different exponential family distributions yield different performances?

---

> ### Author Response · Authors · 2023-11-19
>
> We greatly appreciate the time and effort to review our paper. We hope our following response will resolve most of the concerns.
>
> ## Novelty
>
> While SGD-based approaches are dominating the current CL literature, it is important to note that updating a model with SGD does not come with any theoretical guarantees on performance in CL settings. On the contrary, we highlight a critical aspect of exponential family posteriors, which has been overlooked in the CL community: *they are the only type of posteriors that can theoretically guarantee learning without forgetting while maintaining a constant memory size*. Therefore, we introduce a general MCL framework that leverages the perfect CL ability of statistical models with exponential family posteriors. We effectively address their weak representational power by combining meta-learned neural networks; as a result, we can achieve SOTA performance on a variety of classification, regression, and generation benchmarks. We believe our work provides a fresh perspective on CL, and the other two reviewers also pointed out the novelty as a strength of our approach.
>
>
>
> ## The Choice of Gaussian
>
> While the Fisher-Darmois-Koopman-Pitman theorem is applied to every exponential family distribution, Gaussian distributions are particularly easy to integrate with neural networks thanks to the reparameterization trick [1]. Discrete distributions, such as Bernoulli or categorical distributions, require more sophisticated techniques, such as Gumbel softmax [2]. We also conjecture that the choice of the posterior may not be crucial as long as the meta-learned neural network components are flexible enough. Nonetheless, there may be some cases that can benefit from different types of posteriors, and we leave thorough exploration of them as future work.
>
>
> ---
>
> [1] Kingma and Welling, Auto-Encoding Variational Bayes. ICLR 2014.
>
> [2] Jang et al., Categorical Reparameterization with Gumbel-Softmax. ICLR 2017.

---

### Official Review · Reviewer_y1kN · 2023-10-31

**Soundness:** 3 good
**Presentation:** 3 good
**Contribution:** 3 good
**Rating:** 6
**Confidence:** 4

**Summary:**

The paper proposes a new approach to Meta-Continual Learning (MCL) with the aim to mitigate forgetting in nonstationary streams of data. The novel method, called Sequential Bayesian Meta-Continual Learning (SB-MCL) is presented as a combination of meta-learned neural networks and sequential Bayesian updating. The assumption is that for each CL episode, composed of multiple tasks, a single multidimensional latent variable exists and governs the entire episode. Along the episode, the posterior distribution of latent variable conditional on observations is updated step by step with Bayesian updating to accumulate the entire knowledge of the past. The authors propose the use of the Fisher-Darmois-Koopman-Pitman theorem to show that using distributions from the exponential family enables an update that is resistant to forgetting, while not increasing in time the number of required parameters.  On the opposite, using distributions outside the exponential family in a fixed memory scenario, forgetting becomes inevitable.

The full method uses a neural network encoder (called “learner”) to encode the input as the parameters of an assumed distribution that is used to update the distribution of the latent variable. The posterior distribution is then sampled and another neural network (called “model”) produces an output conditioned on the input and the samples from the posterior.

The encoder and the model are meta-learned across various CL episodes. CL in the inner loop is performed only by Bayesian updating in the latent space, not requiring any gradient computation, enabling to process the stream of data also non-sequentially with the same result. Moreover, this is a way to remove the requirement of computing second-order gradients. The objective of the meta-training optimization is to maximize the log-likelihood of the test set after continually learning from the training stream. Due to the latent variable this function is intractable. The authors introduce a variational distribution of the latent variable (a gaussian distribution) and obtain variational lower bounds both for supervised and unsupervised problems.

SB-MCL generates a family of models that is tested against other MCL competitors in various settings showing the capabilities of the proposal to mitigate catastrophic forgetting.

**Strengths:**

The paper is well written and proposes an innovative and interesting approach to Meta-CL. The proposed idea shifts the focus of the memorization of past experiences from the weight distribution to the latent distribution, removing completely the need of computing gradients during CL phase. The reasons for this choice are well explained and the use of the Fisher-Darmois-Koopman-Pitman theorem enriches the work with a theoretical grounding on how the exponential family can be of fundamental importance to accumulate knowledge in a memory-constrained environment.

The derivation of the variational lower bounds used for continual meta-optimization is also clearly done, overcoming the limitations of previous derivations.

The proposal is tested in a wide range of different tasks, from classification to regression, from generation to image rotation prediction. This confirms the robustness of the idea in different scenarios, improving its significance for the MCL literature.

The proposed method has also a clear and probably impactful improvement from the computational point of view with respect to other models: the possibility to feed the examples in parallel to the learner and the removal of second-order gradients make the method  quite interesting also from a practical point of view.

**Weaknesses:**

One of the main ideas of the paper is the use of distributions from the exponential family to accumulate knowledge from the past without increasing the parameters need. The Fisher-Darmois-Koopman-Pitman theorem implies that distributions from the non-exponential family need sufficient statistics of unbounded dimensions to fully capture the information of the data. While the authors acknowledge that models only based on exponential family update are often too simple, they do not address directly what are the consequences of this simplification in their proposal. Probably the use of a strongly nonlinear model (as a neural network) is sufficient to obtain the same learning power as a model that is not limited to exponential family distribution. In any case it would be interesting to directly address this point.

In a similar way, the sequential Bayesian update with the exponential family has results completely equivalent to having all examples available at the same time. This would solve “by definition” the problem of forgetting. The idea is very interesting, but it should be specified that this is possible due to a simplification assumption on the distribution, and exploring what the consequences of this assumption are. Claiming that this method transforms a CL problem in a underfitting problem should be verified empirically by varying the number of parameters of the neural networks and/or the dimension of the latent space, showing how the memorization ability changes. More importantly this claim is somehow in contradiction with the “losslessness” of the exponential family sequential update, implying that, with the increase of tasks, the number of parameters should increase.

On a higher level, the approach (like any other meta-continual learning framework, in fairness) shifts the forgetting problem from the task to the meta task. Else said, the approach assumes that all meta non-stationarieties are represented at meta-training time. If the meta-test phase characterizes by substantial non-stationarieties the method is bound to fail as it cannot adapt (without incurring once again in catastrophic forgetting at meta level).

**Questions:**

Q1) While the experiments cover a wide array of tasks, it is not clear why the proposed model is not tested in all cases. It would be useful for additional clarity to have the same models tested in all possible cases. If the claim is that the proposed model is the generator of a family of specific models known in the literature, it should be tested anyway, showing if its generality can make an improvement. Is there any technical reason why the proposed general model is not tested in all scenarios?

Q2) As a very small note, the plots with the results are not extremely clear to read, with some lines outside the bounds and some others with similar colours and many overlaps.

---

> ### Author Response · Authors · 2023-11-19
>
> We deeply appreciate the insightful comments. The questions delve into the fundamental aspects of our approach, leading to valuable insights. We provide our answers in the following.
>
> ## The Price of Lossless CL with Exponential Family Posteriors
>
> Compliant with the "no free lunch" theorem, we do pay a price for harnessing the power of exponential family posteriors: the necessity of meta-training. Due to the simplicity of the exponential family, meta-learned neural networks are vital for handling complex and high-dimensional problems. On the other hand, SGD can be applied to standard CL scenarios, although it requires additional techniques to mitigate forgetting. We made this clear in the second paragraph of section 3.3 in the updated draft.
>
> However, as we mentioned in the introduction, MCL is a promising direction toward solving CL. Once we embrace the MCL setting, SB-MCL arises as an attractive solution with minimal downsides. In theory, limiting the inner loop updates to the exponential family posterior should not have a significant impact on the representational capacity, as long as the meta-learned neural networks fulfill their duty as universal function approximators. Our experiments also empirically verify that the meta-learned neural networks are sufficient for SB-MCL to outperform SGD-based MCL approaches.
>
>
>
> ## Underfitting and the Lossless Sequential Update
>
> As neatly summarized in the review, our framework "by definition" produces the same results regardless of how the training data is provided. Our claim about forgetting vs. underfitting was to highlight this fundamental property of our approach, not as an empirical observation. Even in standard multi-task learning settings where all tasks are available throughout training, the performance can degrade as the number of tasks increases, which is often described as underfitting. Since SB-MCL’s performance drop in a CL setting is exactly the same as the performance drop in the corresponding multi-task learning setting with the same tasks, we referred to it as underfitting. More specifically, the sequential Bayesian update is indeed lossless; however, information loss can occur (i) when the learner transforms the raw data to the variational likelihood and (ii) when the model forwards $x$ and $z$. Therefore, underfitting and lossless sequential updates do not contradict each other.
>
> From this perspective, it becomes evident that there would be a positive correlation between the model size and the performance up to a certain point. If we reduce the number of parameters, the performance will surely drop. However, naively increasing the number of parameters does not necessarily lead to a better performance, which is why people do not use a gigantic multi-layer perceptron to solve all kinds of problems. We do need a clever architectural design to further improve the performance, and it would be an exciting topic for future research.
>
>
>
> ## MCL as a Data-Driven Approach to CL
>
> It is true that MCL frameworks transform the challenge of designing a good CL algorithm into the challenge of building a good meta-training set. The meta-training set should be large and diverse enough to cover the meta-test distribution. But is this really a weakness of MCL?
>
> As we mentioned in the introduction, it has been proved that CL is fundamentally an NP-hard problem (Knoblauch et al. 2020). This theoretical result entails a profound message to the CL literature: no matter how meticulously a CL algorithm is designed, there are CL problems that it cannot solve efficiently. This is why we think MCL is a promising direction. Instead of manually designing domain-specific CL algorithms, we can design a general MCL algorithm and collect meta-training datasets for target domains. In this sense, MCL is also better aligned with [The Bitter Lesson](http://www.incompleteideas.net/IncIdeas/BitterLesson.html) by Rich Sutton. It is an argument that general learning algorithms, which leverage more computation and data, have always won over complicated algorithms relying on domain-specific human knowledge in the long run. Therefore, we believe the data-driven aspect of MCL can be considered a strength, rather than a weakness.

---

> > ### Author Response · Authors · 2023-11-19
> >
> > ## Improved Experiments Section
> >
> > We admit that the description of the experimental settings and the presentation of the results were not entirely clear. We have largely improved the overall experiments section in the updated draft, reflecting the comments of reviewer DFZf as well.
> >
> > Since SB-MCL is proposed as a generic framework, there can be multiple instantiations of it depending on the formats of input and output. For image classification and simple regression, GeMCL/PN and ALPaCA are the corresponding instantiations. In these cases, the output formats are simple enough to be handled by GMMs or linear models, and the posterior predictive distribution can be computed analytically. Therefore, there would be no benefit of adding an additional neural output module that introduces additional approximations. Our generic supervised/unsupervised SB-MCL architectures are useful when statistical models are unsuitable for the output, which is the case in the completion, rotation, VAE, and DDPM experiments. In the updated draft, we improve the description of the compared methods, and the results of GeMCL and ALPaCA are also presented under the name SB-MCL to reduce confusion.

---

> > > ### Comment · Reviewer_y1kN · 2023-11-22
> > > **Response to rebuttal**
> > >
> > > First of all, I appreciate the response touch on all points of the review. Please find below some further considerations.
> > >
> > > **The Price of Lossless CL with Exponential Family Posteriors**
> > >
> > > I find the answer quite convincing. The simplification in the latent space should be theoretically compensated by a well-functioning meta-learned neural network that should recover all the representation capabilities. This is similar to what happens in a VAE using a member of the exponential family in the latent space. At the same time, VAE with different distributions exist. Considering that the distributional choice is fundamental to this paper, I would find interesting to direct address the implication of choosing a distribution instead of another for the latent space from the model capabilities perspective. In the end, the reason why the Bayesian update is lossless in the exponential case is exactly the simplicity of these distributions. Are we losing something by choosing this family to accumulate past knowledge?
> > >
> > > **Underfitting and the Lossless Sequential Update**
> > >
> > > In this case, I find the answer only partially solving the issue. Again, the assumption about a single gaussian latent variable that governs the entire episode is an important one and solves the forgetting “by definition”. I agree that the update in the latent space is indeed lossless and that the loss of information can happen at the weights level. Given that this loss of information does not happen in time probably can be called underfitting instead of forgetting. I would still appreciate an empirical validation of this claim due to the important implications it can have on CL: by choosing a large enough model is really possible to solve the CL problem?
> > >
> > > **Improved Experiments**
> > >
> > > I really appreciate the improvements in the section. Now the experiments are more clear and readable.
> > >
> > > To conclude, I am not arguing against MCL. Just commenting on the fact that the risk is that of shifting the issue from the model to the meta-model. Thanks again for your response. I will take this into consideration in the reviewers' discussion and in determining the score.

---

> > > > ### Author Response · Authors · 2023-11-23
> > > >
> > > > We thank the reviewer for the time and effort to provide further feedback. We provide our answers in the following.
> > > >
> > > > ### The Capacity of Exponential Family Posteriors
> > > >
> > > > We find multiple reviewers are concerned about the capacity of exponential family posteriors. The question "Are we losing something by choosing this family to accumulate past knowledge?" seems to be in line with such concern. Despite the simplicity of the exponential family, a high-dimensional exponential family latent variable can contain a huge amount of information since the capacity increases exponentially to the number of dimensions. Therefore, even a single 512-dimensional Gaussian variable can hold a sufficient amount of information to solve our benchmarks.
> > > >
> > > > VAE [1] is a good example that demonstrates the capability of an exponential family latent variable. Another example is GAN [2]. Both of them map a simple Gaussian random variable to real-world images.
> > > >
> > > > While we mentioned the necessity of meta-training as the price to pay, we currently do not see any significant downside of SB-MCL when compared with SGD-based MCL approaches. We think it can be an interesting research topic to find and analyze weaknesses of SB-MCL.
> > > >
> > > >
> > > >
> > > > ### The Diminishing Returns of Increasing the Model Size
> > > >
> > > > Naively increasing the neural network size or the number of dimensions of the latent variable has diminishing returns. It is effective until a certain point, but after that, adding more parameters or dimensions does not lead to a meaningful improvement in performance. For instance, we chose the 512D latent variable because we did not see any significant performance gain beyond that (actually, even 256D was enough).
> > > >
> > > > This is why we emphasized architectural improvement in our initial response and the paper. Simply increasing the number of parameters or latent dimensions would **not** solve CL. We can indeed show that decreasing the number of dimensions (e.g., to 64D or 32D) harms performance. But we are not sure if that would be meaningful since such a trend is universal in all kinds of architecture. If the reviewer still thinks adding such a graph would be helpful, we will certainly add it to the appendix.
> > > >
> > > >
> > > >
> > > > ### Gaussian vs. Other Exponential Family Distributions
> > > >
> > > > Since we can use the reparameterization trick from [1], the Gaussian distribution is most straightforward to integrate with neural networks. Discrete distributions, such as Bernoulli or categorical variables, require a more sophisticated gradient estimation trick [3] introducing additional parameters, such as the temperature parameter in [3]. Therefore, we primarily experimented with the Gaussian distribution in this work and left exploring other exponential family distributions as future work. We currently suspect that naively replacing the Gaussian with some other exponential family distribution would not be helpful, considering the flexibility of meta-learned neural networks. Still, it would be interesting to find cases where other forms of distribution can be beneficial, especially when combined with specific domains or neural network architectures.
> > > >
> > > >
> > > > ---
> > > >
> > > > [1] Kingma and Welling, Auto-Encoding Variational Bayes. ICLR 2024.
> > > >
> > > > [2] Goodfellow et al., Generative adversarial nets. NeurIPS 2014.
> > > >
> > > > [3] Jang et al., Categorical Reparameterization with Gumbel-Softmax. ICLR 2017.

---

### Official Review · Reviewer_DFZf · 2023-11-05

**Soundness:** 3 good
**Presentation:** 1 poor
**Contribution:** 2 fair
**Rating:** 5
**Confidence:** 4

**Summary:**

This paper proposes a new framework for meta-continual learning. The authors propose leveraging the Bayesian learning and specific properties of Gaussian distribution in order to bypass the need for storing large amounts of replay data and use a meta-learned neural network to make the final predictions based on the samples drawn from an episode-specific Gaussian distribution. The authors evaluated their method on a variety of meta learning datasets.

**Strengths:**

The paper is well-organized and structured up to the experiment section. Mathematical derivations are sound and the authors have gone through enough details in order to convey their message.

The idea of leveraging Bayesian learning in the context of continual learning is both interesting and novel.

Figure 1 and 2 nicely describe the whole framework.

**Weaknesses:**

**Using a frozen meta-learned model:** I believe one of the main shortcomings of the proposed method is the fact that the authors propose to freeze the meta-learned network to perform the meta-test.  Freezing the weights extremely limits the plasticity of the framework and I believe Bayesian learning by itself is not enough to address the "different enough" tasks. Also, Using a frozen meta-learned network is analogous to using a pertained model which usually defies the purpose of CL. The end goal of CL is to put the network in scenarios in which truly unseen data are encountered and the network should learn to **adapt** to the new environment while maintaining its previously learned knowledge. I believe freezing is not a good way of maintaining the knowledge and Bayesian learning is not expressive enough to be plastic as it is evident in the qualitative result. The VAE reconstruction results look blurry even for nist-type images.

**Motivation:** I am having a hard time finding a realistic scenario to deploy such a framework. For instance, it seems too restrictive to train VAEs continually while only seeing the data once. At least in the context of generative models, we usually do not operate in such restrictive scenarios.

**Evaluations:** This is tightly related to the first issue. The meta-training tasks are very similar to the meta-testing ones. I believe it is necessary to evaluate the expressiveness of the Bayesian learning approach on more diverse tasks.

**plots:** There are too many plots in Figure 3. Not all of them are informative and some should definitely be moved to the appendix. The message would be better conveyed if we had more tables in the main body instead of a large number of plots in just one figure. The legends are better to be close to the actual plot. It is not easy to match the colors in the top rows to the legend in the bottom row. Also, some of the gray trends (denoted by "standard" in the legend), are out of the plot frame. Especially the one in the bottom right.

**Questions:**

**Q1:** Have the authors tried deploying their framework on more diverse sets of tasks? I am curious to see the limits of expressivity of the Bayesian methods to add plasticity.

---

> ### Author Response · Authors · 2023-11-19
>
> We thank the reviewer for the constructive feedback and for recognizing various strengths in our work. We find that most of the raised concerns are not confined solely to our framework but rather extend to broader meta-learning and MCL settings. In the subsequent discussion, we provide a more comprehensive context within the meta-learning and MCL literature, with the hope of addressing these concerns. In addition, we significantly revised the experiments section following the suggestions.
>
>
> ### Frozen Meta-Learned Components
>
> Freezing some meta-learned components in the inner loop is a well-established technique in both meta-learning and MCL [1, 2, 3, 4, 5, 6, 7]. Moreover, the most important trait of OML [3], our primary SGD-based MCL baseline, is freezing the model’s encoder while training only the two topmost layers in the inner loop. If the encoder is not frozen, OML becomes equivalent to MAML naively applied in MCL settings. The experiments in the original OML paper and our work confirm that the naive MAML performs much worse than OML. This suggests that it is crucial to prevent excessive plasticity, especially in MCL settings, which is contrary to the reviewer’s concern. Even if the subject of inner updates is dramatically simplified, as in the cases of OML and SB-MCL, the meta-learned neural networks are strong enough to compensate for the simplification, demonstrating neural networks’ capabilities as universal function approximators.
>
> In the review, VAE’s blurry reconstructions in the qualitative results were pointed out as evidence for a lack of expressivity in Bayesian learning. However, this is a misinterpretation of the results. Blurry output is a well-known trait of VAE [8], which is orthogonal to Bayesian learning. Our VAE reconstruction results are blurry regardless of the MCL methods (OML, OML-Reptile, and SB-MCL), while the DDPM trained by our SB-MCL can produce incredibly crisp images, which are hard to distinguish from real images.
>
> Lastly, we emphasize that updating neural networks is not the ultimate goal of CL but one possible solution for CL, which is not necessarily ideal. It is straightforward to extend our framework to perform SGD updates of neural networks concurrently in the inner loop. However, updating the neural networks in the inner loop will bring up the forgetting issue, which has been eliminated by our framework, and degrade performance.
>
>
>
> ### Potential Applications of MCL with Generative Models
>
> The integration of MCL with deep generative models holds tremendous potential, unlocking possibilities for various applications that are currently beyond reach. Consider, for instance, text-to-image applications like OpenAI's DALL-E, Adobe's Firefly, or Midjourney that are based on the diffusion model. They significantly expedite the artistic design processes and are already creating substantial economic values. Combining MCL can take this further; artists can supply a few examples of desired results to get a rapidly adapted personalized model. Since the design process often involves gradually adding new assets (e.g., characters, scenes, and objects when creating an animation) built upon the existing assets, continually and rapidly adapting the model to the current project would be especially beneficial.
>
>
>
> ### Meta-Learning Evaluations
>
> As the review pointed out, our meta-training tasks are similar to the meta-testing ones. However, it is one of the most fundamental assumptions of meta-learning [1, 5, 7, 9] and MCL [2, 3, 4], not confined to our approach. In essence, it represents a broader principle applicable to all machine learning paradigms: the training set should cover the test distribution. Even in standard learning scenarios, we generally do not expect a learned model to magically generalize beyond its training set; for example, if we train an image classifier on the images of cats and dogs, it would not perform well on the images of cars or airplanes. This challenge may be related to other topics like open-set classification or out-of-distribution generalization, which are orthogonal to meta-learning and MCL.
>
>
> ### Updated Presentation of the Results
>
> We greatly appreciate the feedback on the plots. The original plot was indeed overly crowded, as we crammed too many results into a small area. We largely updated the experiments section as suggested in the review.
>
> - We added a table in the main text that summarizes the key results.
> - The number of plots is significantly reduced.
> - For each plot, we compare only three methods, which are necessary to deliver our key message: Standard, OML, and SB-MCL.
> - To reduce confusion, the special cases of SB-MCL (PN, GeMCL, and ALPaCA) are now compared under the name SB-MCL, along with the generic SB-MCL architectures for supervised and unsupervised settings.
>
> We believe the updated section is far more concise and clearly conveys the essence of the results.

---

> > ### Author Response · Authors · 2023-11-19
> > **References**
> >
> > [1] Snell et al., Prototypical Networks for Few-Shot Learning. NeurIPS 2017.
> >
> > [2] Banayeeanzade et al., Generative vs. Discriminative: Rethinking The Meta-Continual Learning. NeurIPS 2021.
> >
> > [3] Javed and White, Meta-Learning Representations for Continual Learning. NeurIPS 2019.
> >
> > [4] Beaulieu et al., Learning to Continually Learn. ECAI 2020.
> >
> > [5] Vinyals et al., Matching Networks For One Shot Learning. NeurIPS 2016.
> >
> > [6] Qiao et al. Few-Shot Image Recognition By Predicting Parameters From Activations. CVPR
> > 2018.
> >
> > [7] Chen et al., A Closer Look At Few-Shot Classification. ICLR 2019.
> >
> > [8] Bredell et al., Explicitly Minimizing the Blur Error of Variational Autoencoders. ICLR 2023.
> >
> > [9] Hospedales et al., Meta-Learning in Neural Networks: A Survey. TPAMI 2022.

---

> ### Comment · Reviewer_DFZf · 2023-11-22
> **Response to Authors**
>
> I have read the author's answers and I thank the authors for addressing my concerns and modifying the experiment section. The authors significantly improved their experiment section and explanation of their evaluation settings. I appreciate the removal of excessive figures and the addition of an informative table.
>
> However, I need to clarify some of my previous statements because I believe there is a misunderstanding, especially regarding the task similarities. To me, having similar but challenging tasks is acceptable but when the tasks are too simple to solve (data existing on a low dimensional manifold) like in the NIST-type tasks, and at the same time they are very much similar to each other, then there is a problem since the plasticity and the expressivity of the model cannot be properly tested. With the proposed experiments I cannot really conclude that the exponential models are better than having MLP heads in general or it is just because of the fact that the tasks are trivial. Maybe in some more challenging benchmarks, it would be beneficial to use more complex plastic components. If a proposed method is simple, it should be shown that the simplicity is enough.
>
> Moreover, I encourage the authors to compare their approach with more recent MCL baselines. The only tested baseline is OML which is for 2019.
>
> I thank the authors for pointing out some of the potential applications of their framework however, I believe they are far reached and continual learning is not needed in those applications. The fact that the generative model must generate in an order is different from the fact that it needs to learn all those new design concepts gradually. I still struggle to see the applicability of the framework in real-world problems.
>
> Due to the above-mentioned reasons, I will keep my initial score.

---

> > ### Author Response · Authors · 2023-11-23
> >
> > We deeply appreciate the reviewer for further comments. We are glad to know that the reviewer is satisfied with the updated experiment section. There are still some concerns remaining, but we believe our following answers can address them.
> >
> > ### MCL Baselines
> >
> > Although we have rigorously searched for prior MCL works, we found only three papers that can be classified as MCL: OML [3], ANML [4], and GeMCL [2]. GeMCL has already been tested as a special case of our framework. ANML is a complicated extension of OML that takes much more time and memory to meta-train than OML. According to [the updated OML’s GitHub repository](https://github.com/khurramjaved96/mrcl), its performance is not better than OML after a major bug of OML code was fixed. It is also unclear how to adapt ANML for deep generative models. Therefore, OML was tested as a representative SGD-based MCL method.
> >
> > There are numerous other works that can be classified as online meta-learning (OML) or continual meta-learning (CML) [10, 11, 12, 13, 14, 15, 16, 17]. However, their problem settings are completely different from MCL. While MCL aims to meta-learn how to continually learn, OML and CML aim to incrementally acquire learning ability from a (non-stationary) stream of learning episodes.
> >
> > Compared to OML or CML, MCL has not been researched extensively so far, and we believe our work is an important contribution to this less-explored field of research.
> >
> >
> >
> > ### The Capacity of High-Dimensional Exponential Family Posterior
> >
> > Since the reviewer’s major concern is about the plasticity and expressivity of exponential family posteriors, we would like to note that the dimensionality of the posterior should be taken into consideration. Even exponential family posteriors can hold a huge amount of information if it has hundreds of dimensions, as the capacity increases exponentially to the number of dimensions. While its form is simple, its capacity to hold information should not be underestimated.
> >
> > Moreover, the capability of an exponential family latent variable has been extensively proven in VAEs [18] and GANs [19] where a simple Gaussian variable is mapped to complex images. We hope these concrete examples clear up the concerns about the exponential family.
> >
> >
> >
> > ### Regarding the MCL Benchmarks
> >
> > We would like to emphasize that there is no high-quality large-scale dataset for MCL that is publicly available at the moment. To be a meaningful MCL benchmark, both the number of examples and the number of tasks should be large. The best we could find was the CASIA dataset encompassing 7K classes and 4M examples, and we repurposed it for MCL. To the best of our knowledge, this is the most challenging character recognition dataset, and it is a clear improvement over Omniglot, which has been the standard benchmark for MCL. We respectfully request the reviewer not to devalue a theoretically solid approach because of the use of character recognition datasets.
> >
> >
> >
> > ### Motivation
> >
> > It is unfortunate that the reviewer did not find our example convincing enough. Here, we provide a more general argument, instead of a specific example, about this concern:
> > > I am having a hard time finding a realistic scenario to deploy such a framework.
> >
> > Since our SB-MCL is proposed as a generic MCL framework, it can be applied to almost any MCL setting and model architecture. Meanwhile, MCL can be regarded as a data-driven approach to CL problems that require additional meta-training data. Therefore, if there is a CL scenario that the reviewer considers realistic, our framework can be a potential solution.
> >
> >
> > ---
> >
> > [10] Finn et al., Online meta-learning. ICML 2019.
> >
> > [11] Acar et al., Memory efficient online meta learning. ICML 2021.
> >
> > [12] Yap et al., Addressing catastrophic forgetting in few-shot problems. ICML 2021.
> >
> > [13] Nagabandi et al, Deep online learning via meta-learning: Continual adaptation for model-based RL. ICLR 2019.
> >
> > [14] Jerfel et al., Reconciling meta-learning and continual learning with online mixtures of tasks. NeurIPS 2019.
> >
> > [15] Zhang et al., Variational continual Bayesian meta-learning. NeurIPS 2021.
> >
> > [16] Yao et al., Online structured meta-learning. NeurIPS 2020.
> >
> > [17] Wu et al., Adaptive compositional continual meta-learning. ICML, 2023.
> >
> > [18] Kingma and Welling, Auto-Encoding Variational Bayes. ICLR 2024.
> >
> > [19] Goodfellow et al., Generative adversarial nets. NeurIPS 2014.

---

### Meta-Review · Area_Chair_5gBo · 2023-12-07

**Metareview:**

This paper tries to address the catastropic forgetting issue in continuous learning by converting the model learning process to Bayesian sequential update in a latent space with a Gaussian assumption. The encoding and learner models are meta-learned. Experiments on both supervised learning and generative modeling tasks show better performance than the OML and a few simple baselines.

Multiple reviewers find the proposed combination of meta-learned networks and Bayesian update novel and interesting.

The main concerns from the reviewers are around the exponential family assumption and the resulting simplification. Both reviewer DFZf and y1kN are concerned the capacity of the proposed model with a fixed latent dimensions in the continual learning setting. As commented by reviewer y1kN "shifts the forgetting problem from the task to the meta task". Reviewer DFZf is worried tasks in the benchmarks are too similar to test the limitation of the meta-learned networks. Reviewers' rebuttal does not completely solve these concerns.

**Justification For Why Not Higher Score:**

Shared concerns about the capability of the proposed method in the continuous learning setting. Current experiment results do not providing convincing evidence to relieve it.

**Justification For Why Not Lower Score:**

N/A

---

### Decision · Program_Chairs · 2024-01-16

Reject